# ENCODE, THINK, DECODE: SCALING TEST-TIME REASONING WITH RECURSIVE LATENT THOUGHTS

## ABSTRACT

Most efforts to improve the reasoning capabilities of large language models (LLMs) involve either scaling the number of parameters and the size of training data, or scaling inference computation by letting models generate complex chains of thought. Motivated by interpretability studies showing that the crucial computation required for reasoning tasks is concentrated in a limited range of layers, we introduce Encode–Think–Decode (ETD), a method that enhances the reasoning capabilities of a base model by training it to iterate over a small subset of reasoning-relevant layers during the mid-training stage. ETD amplifies latent reasoning while preserving the original architecture, parameter count, hyperparameters, and training data composition. When iterating on the selected layers at inference time, ETD models yield substantial gains on 17 reasoning benchmarks, including up to +28.4% relative accuracy improvement on GSM8K and up to +36% on MATH with the OLMo-2 1B Base model. We also explore an adaptive depth strategy that adjusts the computation per input token. Our results show that recursive latent reasoning offers a simple and effective path to stronger LLM reasoning.

## 1 INTRODUCTION

Modern language models demonstrate remarkable capabilities in a wide range of reasoning-intensive tasks, including mathematics, programming, commonsense reasoning, and logical puzzles (Brown et al., 2020; Dubey et al., 2024; OpenAI et al., 2023; DeepSeek-AI et al., 2025). The main driver for this progress are scale in both data and parameters, and inference-time techniques such as chain-of-thought prompting.

Initial scaling laws correlated reasoning capabilities to sheer parameter count and training data tokens (Kaplan et al., 2020; Hoffmann et al., 2022; Allen-Zhu & Li, 2024). Ye et al. (2024) refined this picture and argued that depth, not just parameter count, is critical for reasoning: deeper models often outperform shallower ones with the same number of parameters. This perspective aligns with the intuition that reasoning tasks require multi-step, compositional thinking, for which *depth* plays a central role.

Beside scaling data and parameters, the prevalent approach to increasing the reasoning capability of models is by scaling test-time computation. A common approach, known as chain-of-thought (CoT) reasoning (Kojima et al., 2022; Wei et al., 2022), involves prompting or training LLMs to generate intermediate reasoning steps before giving a final answer. This approach emulates human inner monologues and the use of scratchpads, but fails to capture the variability in the amount of non-verbal thought.

An emerging body of interpretability research has also sought to characterize how reasoning is implemented within LLMs. Recent studies suggest that reasoning processes are not uniformly distributed across layers, but instead transition from local, syntactic operations in earlier layers to more global and semantic integration in deeper layers (Elhage et al., 2022; Nanda et al., 2023; Li et al., 2022; Stolfo et al., 2023). Other works highlight the presence of specialized circuits and modular representations that support multi-step inference (Olsson et al., 2022; Singh et al., 2024). These findings suggest that reasoning is not merely a byproduct of scale but is tied to structured computational patterns within the network, motivating architectural modifications that amplify the contribution of reasoning-relevant layers.

Based on these observations, we propose ETD (Encode, Think, Decode), a method to enhance the latent-space reasoning capabilities of existing models by adjusting the effective depth of the network. We identify a range of critical layers for latent reasoning and train it into becoming a recurrent block.

Recursive depth models, also known as looped models, have been mostly studied as a way to improve parameter efficiency (Lan et al., 2019; Bae et al., 2024). Our goal in applying a recursive approach, conversely, is to boost reasoning capabilities by efficiently scaling inference-time computation. There has been work on measuring the effectiveness of recursive-depth models on fairly simple reasoning tasks (Saunshi et al., 2025), and deliberate attempts to improve reasoning via such looping (Geiping et al., 2025). However, these works apply recursion without explicitly targeting the layers most relevant for reasoning within the model.

Rather than training small models from scratch to compare recursive and non-recursive variants, we validate our approach on pretrained open-source models from the OLMo 2 family (OLMo et al., 2024). We re-run their mid-training stage to integrate recursion, but crucially, we do not introduce additional parameters, new data, or changes to the original hyperparameters. This makes our method practical and straightforward to reproduce, as it builds on widely available pretrained models without requiring costly retraining from scratch. To our knowledge, this is the first work to demonstrate that introducing recurrent depth yields significant improvements over modern open-source LLMs.

We demonstrate that our proposed method leads to significant improvements across 17 tasks requiring different types of reasoning. Notably we achieve a relative improvement of 28.4 % and 36% on GSM8K (Cobbe et al., 2021) and MATH (Hendrycks et al., 2021) for the OLMo-2 1B base model.

We also propose how to dynamically set the depth of the model depending on the token. This allows to spend less compute on easy problems and more compute on challenging ones.

The main contributions of the paper are as follows:

- We show that advanced open-source pretrained models can be further enhanced with a recurrent-depth mechanism that requires no additional parameters, training data, or hyper-parameter tuning.

- We demonstrate that ETD provides greater benefits on tasks requiring intensive reasoning, with relative improvements of 28.4% on GSM8K and 36% on MATH for OLMo-2 1B.

- We analyze the impact of iterating over different layers on reasoning performance and introduce a practical recipe for selecting critical layers for latent reasoning.

- We show that performing more latent-space reasoning, i.e. increasing the number of iterations, directly improves performance on reasoning tasks.

- We introduce a mechanism to adaptively determine the number of iterations for each input.

## 2 ON THE ROLES OF LAYERS FOR REASONING

There have been extensive studies on the functional roles of different layers in neural networks. In computer vision, shallow layers are known to capture general features, while deeper layers represent more fine-grained ones (Zeiler & Fergus, 2013; Bau et al., 2017). Similar patterns are also observed in LLMs. For example, Stolfo et al. (2023) show that, when solving simple arithmetic questions, LLMs encode information about operators and operands in mid-sequence early layers, transform this information into intermediate computations in middle layers, and form the representation of the final answer in the last-token middle-to-late layers. Likewise, Zhao et al. (2024) find that, during instruction tuning, early layers capture broad and reusable knowledge, middle layers amplify task-relevant signals, and deeper layers refine these signals into task-specific outputs. More broadly, interpretability studies confirm functional differentiation across layers of varying depths, including in reasoning settings (Yu et al., 2025; Gromov et al., 2024; Shi et al., 2024; Skean et al., 2025).

As information propagates from early to deeper layers, the reasoning process transitions from specific, local, and syntactic information to rich semantic integration. We draw the conclusion that early to middle layers play a critical role in task understanding (Davidson et al., 2025) and knowledge retrieval, while deeper layers are important for higher-level inferences such as those required for mathematical reasoning.

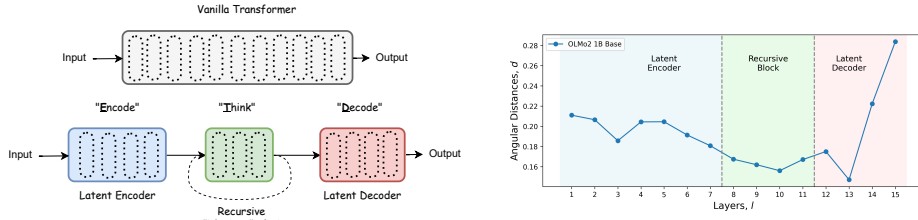

Figure 1: *Left*: Illustration of the proposed architecture (Section 2.1). The latent encoder (blue) maps inputs into latent space, the recursive "thinking" block (green) iteratively refines representations, and the latent decoder (red) maps them back to the output space. Each block consists of a different number of layers. *Right*: Angular distances $d(l, l+1)$ between consecutive layers for OLMo 2 1B base and instruct models. The plot highlights three groups of layers—latent encoder, recursive block, and latent decoder—corresponding to distinct trends in layer-to-layer evolution (Section 2.1).

We therefore break down transformer blocks into three groups (Figure 1): a latent encoder $E$, which embeds the input data into a latent space and retrieves information about mentioned entities, then a core recurrent "thinking" block $T$, a central unit of recurrent computation, that generates latent "thoughts", and finally the latent decoder $D$, which un-embeds from latent space and also contains the prediction head of the model. In practice, the information first goes through layers in the latent encoder $E$, then iterates over the "thinking" block $k$ times, and finally flows through the latent decoder $D$, which returns output tokens. Let's denote different configurations as $N_E$-$N_T$*$k$-$N_D$, e.g. 7-4*2-5 denotes a transformer with 7 layers in the $E$ block, 4 layers in the $T$ block, repeated twice, and 5 layers in the $D$ block.

If the layer-to-layer evolution of representations is given by a residual iteration equation:

$$x^{l+1} = x^l + f(x^l, \theta^l) \qquad (1)$$

where $x^l, \theta^l$ are the input and parameter vectors for layer $l$, and $f(x^l, \theta^l)$ represents the transformation of one multi-head self-attention and MLP layer block (Vaswani et al., 2017), then after $L$ total layers the output is the sum of the input embeddings and the contributions of all the layers:

$$x^L = x^0 + \sum_{l=0}^{N_E-1} f(x^l, \theta^l) + \sum_{j=1}^{k} \sum_{l=N_E}^{N_E+N_T-1} f(x^{l+(j-1)*N_T}, \theta^l) + \sum_{l=N_E+n_T}^{L-1} f(x^{l+(k-1)*N_T}, \theta^l) \quad (2)$$

### 2.1 CHOOSING THE OPTIMAL CONFIGURATION FOR LATENT REASONING

Prior work on related recursive architectures has generally adopted a single predefined partition of layers, without exploring alternatives or analyzing how the choice of split affects performance. Some approaches apply recursion over all internal layers, i.e. employ only a recursive block $T$, (Dehghani et al., 2018; Csordás et al., 2024; Bae et al., 2024; Saunshi et al., 2025), others allocate 1–2 layers each to the $E$ and $D$ blocks (Geiping et al., 2025; Bae et al., 2025; Aleksandrov et al., 2025). In contrast, our work takes the roles of layers into consideration when determining the configuration.

The latent encoder should include enough layers to transform input text into the latent space and retrieve all relevant knowledge, laying the foundation for higher-level semantic analysis and reasoning to happen via a recursive "thinking" block, $T$.

To identify the optimal configuration of layers, we build on the approach of Gromov et al. (2024). They discovered that later layers change the direction of hidden representations less than earlier layers. They used the average angular distance as a criterion for identifying layers to prune. Their experiments show that removing such layers has almost no impact on tasks heavily relying on knowledge retrieval. Despite the low average angular change, however, even moderate pruning of those same layers results in a degradation on reasoning tasks. We build on these insights and use mean angular change to identify reasoning-critical layers to iterate over.

We measure the average change in the direction of the residual stream vector after each layer, and add layers to the latent encoder until the rate of change from layer to layer slows down.

In practice, we compute the average angular distance $d(x(l), x(l+n))$[1], between the input to layer $l$ and the input to layer $l + n$ on the C4 validation set (Raffel et al., 2019). The distance quantifies the degree of update to $x$ resulting from processing between layers $l$ and $l + n$. Figure 1(right) shows the average distances $d(x(l), x(l+1))$ for OLMo-2 1B base and instruct models.

To automatically identify the point, i.e. the layer, at which a curve transitions from a rapid to a gradual decrease, we employ the *Kneedle algorithm* (Satopaa et al., 2011). This method detects "knee" (or "elbow") points in convex, decreasing sequences by analyzing their curvature. Algorithm details are provided in Appendix C. The detected layer index defines the boundary of the latent encoder. For the OLMo-2 1B model, this corresponds to layer 7.

Similarly to the latent encoder, the latent decoder must have sufficient depth to transform representations from the latent space back into the "language" space. To determine the number of layers in the latent decoder, we follow the same procedure as for the latent encoder, but applied in reverse: starting from the final layer of the model and moving backward until reaching the last layer assigned to the latent encoder. For the OLMo-2 1B model, this yields the last 5 layers as the latent decoder. The remaining 4 layers constitute the recursive "thinking" block.

Hence, we set the configuration to 7-4*k-5, i,e. 7 layers in latent encoder, 4 layer in recursive block, and 5 layers in latent decoder respectively, and k is number of iterations. In Figure 1 (right), the rate of change in angular distance decreases around layer 7, stabilizes over the subsequent four layers, and increases again during the final five layers.

Acknowledging that there is no clear single subset of layers solely responsible for reasoning across all models and tasks, we show empirically that our approach selects a split that lies near the performance maximum in the search space across tasks.

## 3 EXPERIMENTAL SETUP

Prior work on recursive-depth models have largely investigated recurrence in training settings that are not representative of modern, fully optimized large-scale LLM pre-training pipelines. We are, however, interested in understanding the impact of recursive "thinking" in realistic scenarios, and therefore apply them on open-source models trained following best practices in architecture, training recipe, and pretraining data mixtures. We base our study on the OLMo 2 family of models (OLMo et al., 2024), focusing specifically on the base configurations. For fair comparison, our ETD models use the same number of parameters, datasets, and hyperparameters as the baseline non-recursive model.

### 3.1 TRAINING PIPELINE

OLMo 2 is a family of LLMs with open artifacts including intermediate and final checkpoints, training data, code, and recipes for 1B, 7B and 13B scale models, both pre-trained and post-trained. As a compromise between experimental agility and model power, we focus on 1B parameter model. We integrate ETD into the existing training pipeline without introducing additional training steps or data. This requires access to the model weights, training data, and hyperparameters to evaluate the impact of ETD in a controlled and isolated manner.

Following recent advances in curriculum learning (Blakeney et al., 2024; Ibrahim et al., 2024) OLMo 2 base models are trained in two stages. The first (pretraining) stage is the longest ($\geq 90\%$ training FLOPs), and uses mostly web-sourced data. The second stage, which is referred to as mid-training (5-10 % of training FLOPs), upsamples the highest-quality web documents and curated non-web sources. The purpose of this mixture is to imbue the model with reasoning skills and provide focused exposure to STEM references and high quality text.

We evaluate the EDT approach by integrating it into the mid-training stage which uses only 1.25% of the total pretraining tokens.[2] In our experiments, we initialize the model with the weights after the first stage training and run the mid-training with ETD approach for each configuration separately. OLMo et al. (2024) perform mid-training with three random orders, then average the resulting mod-

---

[1]We explain the details of computing angular distance in Appendix A

[2]For the OLMo-2 1B model, stage-1 pretraining uses $4 \times 10^{12}$ tokens, while stage-2 uses $5 \times 10^{10}$ tokens.

els. In our setup, we train with one data configuration and compare it to the standard model trained with the same configuration. Since our experiments adopt the same data mixtures and configurations, we direct readers to OLMo et al. (2024) for a comprehensive description of the training pipeline.

## 3.2 EVALUATION BENCHMARKS

To capture broad conceptual nature of reasoning, we consider 17 real-world benchmarks grouped into six categories, ordered along a spectrum from less to more reasoning intensive tasks, i.e. from factual recall to systematic symbolic reasoning: factual knowledge, reading comprehension, commonsense reasoning, multidisciplinary Reasoning, BIG-Bench Hard (BBH), and mathematical reasoning. This progression reflects increasing reliance on reasoning rather than memorization. We provide the task categories with the corresponding benchmarks in Table 1. Details

Table 1: Evaluation benchmarks grouped into six categories, listed in order of increasing reasoning intensity from top to bottom.

| Category | Benchmarks |
|---|---|
| Factual Knowledge | TriviaQA, NaturalQuestions |
| Reading Comprehension | BoolQ, OpenBookQA, DROP |
| Commonsense Reasoning | CommonSenseQA, HellaSwag SocialQA, WinoGrande |
| Multi-Disciplinary Reasoning | ARC-Easy, ARC-Challenge, MMLU, MMLU-Pro, AGIEval-English |
| BIG-Bench Hard | BBH [3] |
| Mathematical Reasoning | GSM8K, MATH |

with the motivation for each task category are provided in Appendix B. We evaluate the model using OLMES (Gu et al., 2024), a standardized evaluation suite and toolkit.

Table 2: Results of the Encode–Think–Decode (ETD) method with varying numbers of iterations over recursive "thinking" blocks, compared to the OLMo 2 1B baseline. Reported metrics include accuracy (Acc.) and relative improvement ($\Delta$, in %) with respect to the baseline, for each of six task categories (as defined in Sec. 3.2). Parameter counts denote the number of distinct layers, while FLOPs correspond to the number of effective forward-pass layers.

| Model | Params/FLOPs | Factual Knowledge | | Reading Comprehension | | Commonsense Reasoning | | Multi-Disciplinary Reasoning | | BBH | | Math. Reasoning | |
|---|---|---|---|---|---|---|---|---|---|---|---|---|---|
| | | Acc. | $\Delta$(%) | Acc. | $\Delta$(%) | Acc. | $\Delta$(%) | Acc. | $\Delta$(%) | Acc. | $\Delta$(%) | Acc. | $\Delta$(%) |
| OLMo 2 (k=1) | 16 / 16 | 37.55 | - | 52.19 | - | 65.29 | - | 45 | - | 31.8 | - | 24.31 | - |
| ETD (k=2) | 16 / 20 | 38.1 | (+1.5%) | 56.14 | (+7.6%) | 66.74 | (+2.2%) | 48.41 | (+7.6%) | 31.67 | (-0.4%) | 28.27 | (+16.3%) |
| ETD (k=3) | 16 / 24 | 37.55 | (0%) | 56.07 | (+7.4%) | 67.75 | (+3.77%) | 49.55 | (+10.1%) | 32.62 | (+2.6%) | 30.29 | (+24.6%) |
| ETD (k=4) | 16 / 28 | 37.74 | (0%) | 57.76 | (+10.7%) | 68.16 | (+4.4%) | 50.18 | (+11.5%) | 33.01 | (+3.8%) | 29.62 | (+21.8%) |
| ETD (k=5) | 16 / 32 | **38.23** | (+1.8%) | **58.5** | (+12.1%) | **68.41** | (+4.8%) | **50.58** | (+12.4%) | **33.49** | (+5.3%) | **30.45** | (+25.3%) |

## 4 EVALUATING RECURSIVE "THINKING" BLOCKS

All results are obtained using the training pipeline described in Section 3.1, with the only modification being the configuration $N_E$-$N_T$*$k$-$N_D$. Here, $N_E$, $N_D$, and $N_T$ denote the number of layers in the latent encoder and decoder, and the recursive block, and $k$ is the number of iterations. Since our objective is to evaluate the model's reasoning abilities, we focus on reasoning-oriented tasks as defined in Section 3.2. Because we deal with the same architecture while changing only the number of layers, we report the number of parameters in terms of distinct layers, $N_E+N_T+N_D$, and the number of FLOPs in terms of forward passes through layers, $N_E+N_T*k+N_D$.

### 4.1 PERFORMANCE GAINS FROM ITERATING OVER "THINKING" BLOCKS

We begin by examining the first two rows of Table 2, which report results for the baseline and the recursive model with two iterations, corresponding to the 7–4*2–5 configuration. Notice that the OLMo 2 1B baseline is equivalent to the ETD model with $k$=1. Results show that performance either remains stable or improves, with notable gains in several categories. The largest improvement is observed on Mathematical Reasoning tasks, with an average relative increase of 16.3%. A breakdown in Table 3 confirms that both GSM8K and MATH benefit from two iterations of the ETD

---

[3]BBH, a collection of 23 diverse tasks, serves as a cross-cutting benchmark for compositional reasoning that does not fit neatly into the other categories. More details in Appendix B

approach. Additional gains appear in Commonsense Reasoning (+2.2%), Reading Comprehension (+7.6%), and Multi-Disciplinary Reasoning (+7.6%). In contrast, tasks in the Factual Knowledge and BIG-Bench Hard categories exhibit at most marginal benefits from a single additional iteration.

To further assess the effect of recursive processing, we train ETD with varying numbers of iterations, with results summarized in Table 2. Performance generally improves as the number of iterations $k$ increases with one notable exception: the Factual Knowledge category shows negligible improvement. As discussed in Section 3.2, these tasks rely mainly on memorization rather than reasoning. In contrast, the largest gains occur in reasoning-intensive tasks, most notably in Mathematical Reasoning, with breakdowns shown in Table 3.

Table 3: Results of the ETD method with varying numbers of iterations. Reported metrics include accuracy (Acc.) and relative improvement ($\Delta$, in %) with respect to the baseline on the mathematical reasoning tasks, GSM8K and MATH.

| Model | Params/FLOPs | GSM8K | | MATH | |
|---|---|---|---|---|---|
| | | Acc. | $\Delta(\%)$ | Acc. | $\Delta(\%)$ |
| OLMo 2 (k=1) | 16 / 16 | 44.05 | - | 4.57 | - |
| ETD (k=2) | 16 / 20 | 51.10 | (+16.01%) | 5.45 | (+19.22%) |
| ETD (k=3) | 16 / 24 | 54.36 | (+23.41%) | **6.22** | **(+36.04%)** |
| ETD (k=4) | 16 / 28 | 55.50 | (+25.99%) | 3.73 | (-18.28%) |
| ETD (k=5) | 16 / 32 | **56.56** | **(+28.4%)** | 4.33 | (-5.17%) |

These results demonstrate that the ETD approach—by iterating over reasoning-relevant layers—substantially enhances the non-recursive baseline, yielding relative improvements of +28.4% on GSM8K and +36% on MATH. Moreover, the minimal gains on memorization tasks further validate our approach from Section 2 for identifying layers specialized in reasoning.

As noted earlier, ETD with $k=2$ iterations shows no improvement on BIG-Bench Hard (BBH) tasks. However, performance begins to increase with $k=3$ and continues to improve with additional iterations. These observations highlight that performance as a function of iterations exhibits different trends across tasks. For some tasks (e.g., Social IQa, ARC-Challenge, MMLU), performance rises rapidly with 2–3 iterations, after which the rate of improvement slows. For others (e.g., DROP, MMLU-Pro, GSM8K), gains continue steadily with each additional iteration. In rare cases, the best performance is not achieved at the maximum depth, as observed for MATH. Detailed results for all 17 tasks are provided in Appendix F.

Overall, these findings indicate that allocating more resources to generating latent "thought" before decoding—that is, by performing additional iterations over the "thinking" blocks—systematically enhances performance on reasoning-oriented tasks. The diverse performance trends across tasks highlight the opportunity to explore input-dependent, adaptive-depth recursive methods, which we investigate in Section 5.

Our results empirically demonstrate that the methodology described in Section 2 enables the selection of configurations that enhance the model's reasoning capabilities. Notably, the experiments in the following sections show that it lies near the performance maximum in the search space across tasks.

## 4.2 COMPARISON WITH ALTERNATIVE RECURSIVE FRAMEWORKS

Prior work on recursive LLMs typically applies recursion either across all layers (Dehghani et al., 2018; Csordás et al., 2024; Bae et al., 2024; Saunshi et al., 2025) or across middle layers while preserving a few initial and final layers (Geiping et al., 2025; Bae et al., 2025; Aleksandrov et al., 2025). For a fair comparison, we train models using both strategies: (i) looping over all layers, and (ii) a 2–12*2–2 configuration, which repeats the middle 12 layers while keeping two layers at the beginning and end fixed. We compare these baselines to our selective looping configuration under a constant FLOP budget, with results shown in Table 4.

Our approach consistently outperforms these alternatives under equal compute. For example, the 2–12*2–2 setup is FLOP-equivalent to our 7–4*4–5 configuration, yet yields lower accuracy. Moreover, to match or exceed the performance of alternative strategies, our method typically requires fewer FLOPs—often only three iterations are sufficient. We also want to note that $N_E=N_D=0$ configuration in Table 4, is the closest analogue to Coconut (Hao et al., 2024).

Table 4: Results with recursive baselines

| Model | Params/ FLOPs | Factual Knowledge | Reading Comprehension | Commonsense Reasoning | Multi-Disciplinary Reasoning | BBH | Math. Reasoning |
|---|---|---|---|---|---|---|---|
| OLMo 2 | 16 / 16 | 37.55 | 52.19 | 65.29 | 45 | 31.8 | 24.31 |
| 2-12*2-2 | 16 / 28 | 37.7 | 56.44 | 67.73 | 47.58 | 32.30 | 29.27 |
| ETD (k=4) | 16 / 28 | **37.74** | **57.76** | **68.16** | **50.18** | **33.01** | **29.62** |
| 0-16*2-0 | 16 / 32 | 37.35 | 53.58 | 64.7 | 45.24 | 30.59 | 24.99 |
| ETD (k=5) | 16 / 32 | **38.23** | **58.5** | **68.41** | **50.58** | **33.49** | **30.45** |

Figure 2: Results of the ETD method when varying the subset of layers in the recursive block. We report accuracy (Acc.) when increasing the size of the latent encoder $N_E$ from 1 to 11 in steps of 2, for each of 6 task categories (as defined in Sec. 3.2). The orange line marks selected configuration.

### 4.3 HOW DOES THE CHOICE OF RECURSIVE LAYERS CHANGE PERFORMANCE?

To further examine the impact of recursive layer choice, we fix the recursive "thinking" block size and vary its starting position from layer 2 to 12 in steps of 2, which is equivalent to increasing the size of the latent encoder $N_E$ from 1 to 11 in steps of 2. An intriguing observation is that the optimal configuration slightly varies depending on the specific category of tasks. The results in Figure 2 show that the 7-4*2-5 configuration achieves the best overall performance on reasoning-intensive task, particularly mathematical reasoning. Detailed results are in Table 7 in Appendix E. A close alternative is 5-4*2-7, which performs comparably on most tasks but falls short in mathematics. Performance on Factual Knowledge tasks is stable across configurations, which aligns with the intuition discussed earlier. Interestingly, for reading comprehension, the 3-4*2-9 configuration performs best. This block of layers (4-7) overlaps with layers just before the identified "thinking" block (8-11), aligning with our earlier intuition that early-to-middle layers are important for context understanding. These findings are consistent with our layer-role analysis, though further investigation is needed to establish stronger causal links.

### 4.4 HOW DOES THE SIZE OF RECURSIVE "THINKING" BLOCK CHANGE THE PERFORMANCE?

To ensure a controlled comparison, we vary the size of the recursive block by symmetrically adding or removing layers around the original 7–4×2–5 configuration, keeping its center fixed while changing its extent. Figure 3 shows that performance increases as more layers are included in the recursive block up to a point, after which it begins to decline. Notably, for mathematical reasoning, and even under the same FLOP budget, looping more times over a compact set of layers (7–4×k–5) outperforms looping fewer times over a larger set of layers. This suggests that the placement and structure of the recursive computation are key drivers of performance, not just the amount of extra compute[4].

### 4.5 COMPARISON WITH LARGER MODEL WITH SAME EFFECTIVE DEPTH

We perform an iso-FLOPs comparison by matching the effective depth of ETD with k=2. The 7-4*2-5 configuration has an effective depth of 22, so we construct a non-recurrent baseline with the same budget by stacking the 4-layer block twice—yielding a 7–8×1–5 configuration that mimics two iterations without recurrence. Both configurations perform identically before mid-training. However, results in Table 5 show that the larger iso-FLOPs model underperforms both the original

---

[4]Detailed results are in in Appendix G.

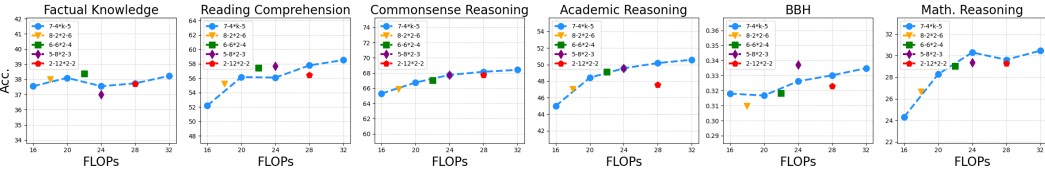

Figure 3: Results of the ETD method when varying the number of layers in the recursive block. We report accuracy (Acc.) when changing the size of the latent encoder $D$ between 2,4,6,8, and 12, for each of 6 task categories (as defined in Sec. 3.2). Each color represents different configuration of $N_E$-$N_T$*k-$N_D$.

non-recurrent baseline and the ETD (k=2) model, highlighting the importance of reusing reasoning-critical layers rather than expanding the network.

Table 5: Results with larger model and same FLOPs

| Model | Params/ FLOPs | Factual Knowledge | Reading Comprehension | Commonsense Reasoning | Multi-Disciplinary Reasoning | BBH | Math. Reasoning |
|---|---|---|---|---|---|---|---|
| OLMo 2 | 16 / 16 | 37.55 | 52.19 | 65.29 | 45 | 31.8 | 24.31 |
| 7-8-5 | 20 / 20 | 31.78 | 52.03 | 62.45 | 44.42 | 30.21 | 21.68 |
| 7-4*2-5 | 16 / 20 | **38.1** | **56.14** | **66.74** | **48.41** | **31.67** | **28.27** |

## 4.6 SCALING FROM 1B PARAMETERS TO 7B PARAMETERS

We extend our experiments from the 1B model to the 7B model. Applying the configuration selection procedure from Section 2.1 yields the 16–10*2–6 layer assignment, which we train using the same mid-training ETD integration described in Section 3.1. The 7B experiments follow the same qualitative trends observed at 1B scale: as shown in Table 6, ETD consistently improves mathematical reasoning perfor-

Table 6: Results of the ETD method on OLMo-2 7B base model. Reported metrics are accuracy (Acc.) and relative improvement ($\Delta$, in %) with respect to the baseline.

| | | GSM8K | | MATH | |
|---|---|---|---|---|---|
| Model | Params/FLOPs | Acc. | $\Delta$(%) | Acc. | $\Delta$(%) |
| OLMo 2 7B (k=1) | 32 / 32 | 66.18 | - | 17.07 | - |
| ETD (k=2) | 32 / 42 | 67.02 | (+1.29%) | 18.26 | (+6.38%) |

mance, while gains on other task categories are less pronounced (see Appendix H). We note that mid-training of both 1B and 7B models uses the same amount of data, meaning that 1B was exposed to more data per parameter.

## 5 ADAPTIVE TEST-TIME SCALING

We observed significant improvements of iterating over recursive blocks. The general trend is that the model benefits from more iterations. However, different problems demand different levels of reasoning effort: not all tokens or sequences require the same number of iterations to reach an accurate prediction, and in some cases the marginal benefit of additional iterations may not justify the extra computation. Adaptive computation (Bengio et al., 2013; 2015) is often used for efficiency by early-exiting on simpler tokens (Elhoushi et al., 2024). In contrast, our goal is to adaptively allocate computation at test time to enhance reasoning capability, rather than to reduce cost.

### 5.1 METHODOLOGY

In our architecture of the form $E \rightarrow T*k \rightarrow D$, instead of fixing the number of recursive iterations $k$, we adopt the Adaptive Computation Time (ACT) mechanism (Graves, 2016), allowing each token to dynamically determine how many applications of the recursive block $T$ are necessary. A *router* evaluates the hidden state after each iteration and decides whether further computation is required. This enables allocating more steps to tokens that demand deeper reasoning, while those not meeting the selection criteria bypass further processing and retain their previous representation.

At each iteration $t$, after computing the hidden representation $h_t$ with the recursive block, a *router* predicts a halting values $w_t \in (0, 1)$ for each token. These values are accumulated across iterations:

$$H_t = \sum_{j=1}^{t} w_j. \tag{3}$$

Computation for a token is stopped once $H_t \geq 1 - \epsilon$, with $\epsilon$ is a small constant (e.g. 0.01). Intuitively, each $w_t$ represents the confidence of the latent "thought", as produced by the recursive block $T$. Until sufficient confidence is accumulated, the latent "thought" state continues to be updated. The final representation passed to $D$ is the output of "thinking" block $T$ after final iteration. [5]

Despite its simplicity, this design proved effective in practice. Compared to a fixed-depth design, ACT introduces per-token dynamic depth, enabling more efficient and adaptive use of the recursive block. Full details are provided in Appendix D.

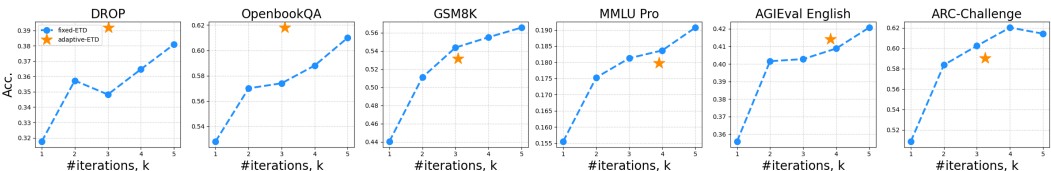

Figure 4: Results of fixed-depth ETD with varying numbers of recursive "thinking" iterations compared to adaptive-depth ETD. For fixed-depth ETD, we report accuracy (Acc.) at each iteration count. For adaptive-depth ETD, we report accuracy and the average number of iterations per task.

## 5.2 RESULTS

We outlined the difference in architecture between fixed- and adaptive-depth approaches, while we follow the same training pipeline discussed in Section 3.1. Figure 4 reports the performance of fixed-depth ETD and adaptive-depth ETD, together with the average number of loops per task.[6]

From Figure 4, we make three key observations. First, this exploratory approach in the direction of adaptive test-time compute approach shows clear improvement over baseline with no recursive iterations. Second, looking at the performance on DROP and OpenbookQA, both of which are reading comprehension tasks, we see that adaptive-depth ETD outperforms the ETD with fixed $k$=5 iterations. Moreover, it also achieves this with fewer iterations on average. Third, for the remaining tasks, adaptive-depth ETD follows the empirical accuracy–iteration tradeoff of the fixed-depth baselines. In particular, its accuracy matches the trend observed for increasing iteration counts, suggesting that performance is well-aligned with its average effective depth. Notably, in these tasks, the adaptive method halts additional iterations once further computation yields only marginal gains.

## 6 RELATED WORK

**Recursive architectures** Recurrence has long been a foundational concept, from RNNs to efforts to incorporate it into transformers. In transformers, recurrence has been explored by iteratively refining representations across all tokens in parallel (Dehghani et al., 2018; Lan et al., 2019), and applied to algorithmic tasks such as arithmetic (Schwarzschild et al., 2021; Bansal et al., 2022; Bear et al., 2024; McLeish et al., 2024). Other works offered theoretical and small-scale analyses of looped transformers (Giannou et al., 2023; Gatmiry et al., 2024; Yang et al., 2023; Fan et al., 2024).

Beyond fully recurrent-depth architectures, several hybrid designs have also been proposed, including latent sub-networks (Li et al., 2020), Mixture-of-Experts structures (Tan et al., 2023; Csordás et al., 2024), and dynamic weight-tying (Hay & Wolf, 2024; Liu et al., 2024b). The major motivation of many works mentioned above was inspired by efficiency based on utilizing shared parameters.

---

[5]We also tried to follow Graves (2016) to represent final representation as the weighted mixture of the outputs after each iteration, but found it less effective.

[6]We selected these tasks because they exhibit the largest relative gains from the recursive approach. See Appendix F for results on the six tasks with the highest relative improvement of ETD ($k$=5) over baseline.

**Latent Reasoning** Chain-of-thought prompting has been a central focus in recent studies of reasoning (Merrill & Sabharwal, 2024; Feng et al., 2023; Li et al., 2024). In contrast, our proposal follows the alternative line of latent reasoning, where reasoning unfolds in the model's hidden representations rather than explicit textual traces. Related efforts on learning to reason in continuous spaces include Hao et al. (2024); Cheng & Durme (2024); Liu et al. (2024a); Geiping et al. (2025); Saunshi et al. (2025). Chen & Zou (2024); Ye et al. (2024); Petty et al. (2023) have shown the importance of model depth for reasoning. Further analysis on Coconut (Hao et al., 2024), show that continuous thought vector is a superposition state that encodes multiple search frontiers simultaneously (Zhu et al., 2025b;a). We step further showing that larger depth leads to reasoning improvements also when it is achieved via looping, without increasing the number of parameters.

**Adaptive Computation** Dynamic compute allocation has been shown to substantially reduce training and inference costs, spanning from early neural networks (Bengio et al., 2015; Huang et al., 2016; Teerapittayanon et al., 2016; Panda et al., 2015) to LLMs (Hou et al., 2020; Elbayad et al., 2019; Fedus et al., 2021; Bae et al., 2023; Elhoushi et al., 2024). A prominent line of work, early exiting, learns to terminate computation on "easy" inputs by skipping subsequent layers (Elbayad et al., 2019; Schuster et al., 2022; Bae et al., 2023; Elhoushi et al., 2024). Adaptive depth can be also formulated as a routing problem: each layer's router selects a subset of tokens for full computation while others bypass the layer, enabling token-level conditional compute (Raposo et al., 2024; Luo et al., 2024). Extending this idea, Bae et al. (2025) applied conditional routing to recursive transformers, but restricted recursion to a small, fixed maximum of three iterations.

**Key Differences from Prior Work** Our approach differs from prior work in several important ways. First, most recursive-depth methods have been studied primarily as a means of improving parameter efficiency (Lan et al., 2019; Bae et al., 2024), i.e., reducing parameter count while maintaining performance, whereas our focus is on enhancing reasoning capability. Second, to our knowledge, we are the first to propose a recursive approach guided by interpretability: rather than choosing the recursive configuration heuristically, we iterate specifically over layers critical for reasoning. Third, our method is simple and requires no additional components such as extra latent states for recursive blocks and very large of number of iterations (Geiping et al., 2025), LoRA adapters (Bae et al., 2024), regularization terms (Saunshi et al., 2025), or input injections (Aleksandrov et al., 2025). Unlike methods such as Coconut (Hao et al., 2024), which introduce a separate language and latent mode, and multi-stage training, ETD preserves the standard forward pass and applies recurrence only to a small reasoning-critical block—yielding stronger reasoning gains. Fourth, unlike most prior work that evaluated recurrence under simplified setups, we show that recursive depth improves advanced open-source models trained with state-of-the-art practices in architecture, training recipes, and pretraining mixtures, validating our approach extensively on real-world reasoning tasks. Speaking of adaptive-depth recursive model, in our formulation we advocate for open-ended test-time compute scaling: after each iteration, the model should autonomously decide whether to continue or halt, without being constrained by a predefined cap (Bae et al., 2025).

## 7 CONCLUSIONS

We introduced *Encode–Think–Decode* (ETD), a paradigm that enhances the reasoning abilities of LLMs by performing latent-space reasoning. Unlike approaches that depend on scaling model size or externalizing reasoning through CoT prompting, ETD amplifies reasoning-relevant computations within the model itself, without altering its architecture, parameters, data, or hyperparameters. Across 17 benchmarks, ETD consistently improved performance, with substantial gains on reasoning-intensive tasks such as GSM8K and MATH. Our analysis underscores the importance of iterating over deeper, reasoning-relevant layers, and adaptive depth strategies further show how ETD can dynamically allocate compute based on task difficulty.

Overall, recursive latent reasoning emerges as a simple, effective, and broadly applicable approach for strengthening reasoning. By integrating interpretability insights with recursive computation, ETD illustrates how leveraging depth and structure can advance reasoning in language models.

## ETHICS STATEMENT

Our study focuses on methodological contributions for enhancing reasoning in large language models and relies exclusively on publicly available datasets and open-source pretrained models. We do not introduce new data, nor do we involve human subjects. We do not foresee direct societal risks beyond those already associated with language models. At the same time, we hope that improving the reasoning ability of models can lead to safer and more reliable applications by reducing errors in reasoning-intensive domains.

## REPRODUCIBILITY STATEMENT

We build on openly released models, which provide full access to weights, data mixtures, and training recipes. Our modifications involve only the mid-training stage, where we re-run training with the same data and hyperparameters, adding recursive iterations without introducing new parameters or datasets. All evaluations use widely available benchmarks. We report full configuration details, including recursive block structure and iteration counts in the main text and appendices. These choices ensure that our results can be reproduced by others with access to the training pipeline and publicly available evaluation benchmarks.

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

## A  COMPUTING ANGULAR DISTANCE

Elaborating on the computation of angular distance in Section 2.1, the angular distance for a single sequence of length $T$ is defined as

$$d\left(x^{(\ell)}, x^{(\ell+n)}\right) = \frac{1}{\pi} \arccos\left(\frac{x_T^{(\ell)} \cdot x_T^{(\ell+n)}}{\|x_T^{(\ell)}\| \, \|x_T^{(\ell+n)}\|}\right),$$

where the inner product is taken over the hidden dimension of the model for the last token $T$ of the sequence, $\|\cdot\|$ denotes the $L^2$ norm, and the factor $1/\pi$ normalizes the distance to $[0, 1]$. We average this distance over 10,000 examples to obtain a stable estimate. We focus on the final token since, under a causal attention mask, its embedding is the only one that depends on the entire sequence. We use the same definition of angular distance as Gromov et al. (2024).

## B  DETAILED EVALUATION BENCHMARKS

To capture broad conceptual nature of reasoning, we consider 17 real-world benchmarks grouped into six categories, ordered along a spectrum from less to more reasoning intensive tasks, i.e. from factual recall to systematic symbolic reasoning: factual knowledge, reading comprehension, commonsense reasoning, multi-disciplinary Reasoning, BIG-Bench Hard (BBH), and mathematical reasoning. This progression reflects increasing reliance on reasoning rather than memorization.

- **Factual Knowledge:** Tasks that test the model's ability to recall information without additional context, thus primarily measuring memorization. We include TriviaQA (Joshi et al., 2017) and NaturalQuestions (Kwiatkowski et al., 2019).

- **Reading Comprehension:** Tasks requiring the model to infer answers from a given passage, involving text understanding and light reasoning (e.g., multi-hop). Benchmarks include BoolQ (Clark et al., 2019), OpenBookQA (Mihaylov et al., 2018), and DROP (Dua et al., 2019).

- **Commonsense Reasoning:** Tasks that evaluate human-like capacity to make assumptions and inferences about the nature and characteristics of everyday scenarios, including CommonSenseQA (Talmor et al., 2019), HellaSwag (Zellers et al., 2019), SocialQA (Sap et al., 2019), WinoGrande (Sakaguchi et al., 2021).

- **Multi-Disciplinary Reasoning:** Benchmarks testing both factual knowledge and reasoning across broad academic and multi-disciplinary domains. We include ARC-Easy and ARC-Challenge (Clark et al., 2018), MMLU (Hendrycks et al., 2020), MMLU-Pro (Wang et al., 2024), and AGIEval-English (Zhong et al., 2023).

- **BIG-Bench Hard (BBH):** A collection of 23 diverse tasks spanning math, logic puzzles, symbolic and social reasoning (Suzgun et al., 2022). Many tasks are synthetic, and BBH serves as a cross-cutting benchmark for compositional reasoning that does not fit neatly into the other categories.

- **Mathematical Reasoning:** We finally test the model on solve math word problem benchmarks to evaluate systematic reasoning and symbolic manipulation, represented by GSM8K (Cobbe et al., 2021) and MATH (Hendrycks et al., 2021).

## C  ALGORITHM FOR CHOOSING THE OPTIMAL CONFIGURATION

To automatically identify the point at which a curve transitions from a rapid to a gradual decrease, we employ the *Kneedle algorithm* (Satopaa et al., 2011). The difference function $D_i$ is then evaluated on $(x, \tilde{y}(x))$, providing a smooth approximation that avoids spurious local variations.

Formally, let the curve be represented as a sequence of points:

$$\mathcal{C} = \{(x_i, y_i)\}_{i=0}^n,$$

where $x$ corresponds to the layer index $l$ and $y$ to the angular distance $d(l, l+1)$. The key steps underlying Kneedle Algorithm are:

1. Smooth and normalize the data into $[0,1]^2$: $(\hat{x}_i, \hat{y}_i)$.

2. Compute the deviation $D_i = \hat{y}_i - (1 - \hat{x}_i)$ from the diagonal.

3. Identify local maxima of the difference curve as candidate knees.

4. Apply a threshold-based rule (with sensitivity parameter $S$) to declare knees when the difference drops below threshold.

To improve robustness against noise, we apply a polynomial interpolation of degree 2 to the data:

$$\tilde{y}(x) = a_0 + a_1 x + a_2 x^2,$$

fitted via least squares. This provides a smooth approximation that avoids spurious local variations.

The details of Kneedle Algorithm can be summarized as follows:

1. Normalization: Scale both axes to $[0,1]$:

$$\hat{x}_i = \frac{x_i - \min(x)}{\max(x) - \min(x)}, \qquad \hat{y}_i = \frac{y_i - \min(y)}{\max(y) - \min(y)}.$$

2. Difference curve: Compute the deviation between the normalized curve and the diagonal $y = 1 - \hat{x}$:

$$D_i = \hat{y}_i - (1 - \hat{x}_i).$$

3. Local maxima: Candidate knees are local maxima of $D_i$, i.e.

$$D_{i-1} < D_i \quad \wedge \quad D_{i+1} < D_i.$$

4. Threshold rule: For each local maximum, define a threshold

$$T_i = D_i - S \cdot \Delta_x, \quad \Delta_x = \tfrac{1}{n-1} \sum_{j=1}^{n-1} (\hat{x}_{j+1} - \hat{x}_j),$$

where $S > 0$ is a sensitivity parameter. A knee is declared at $i^*$ if $D_j < T_i$ for some $j > i$ before the next local maximum is reached.

We run the above procedure using the `KneeLocator` package:

```
kneedle = KneeLocator(
    x, y,
    curve='convex',
    direction='decreasing',
    interp_method='polynomial',
    polynomial_degree=2,
    online=True
)
```

The returned index

$$i^* = \texttt{kneedle.knee}$$

is taken as the transition point from steep to gradual decline.

## D   DETAILS ON ADAPTIVE-DEPTH ETD TRAINING

In Section 5, we introduce the mechanism that allows the model to adaptively determine the number of recursive iterations per input token—referred to as adaptive-depth ETD. This subsection provides full implementation details covering the architecture, training, and inference procedure.

**Architecture.**   We keep the general architecture of the model the same and add a lightweight router. The router is implemented as a linear projection of the hidden state followed by a sigmoid activation. The input to the router is the hidden representation that is output by the recursive T block, and the output of the router is the halting value between 0 and 1. The router is randomly initialized, i.e. we do not use the insights from fixed-depth ETD to set some priors for the router.

**Training stage.** Adaptive-depth ETD undergoes mid-training in the same way as fixed-depth ETD. We train the router to learn how to allocate resources, i.e. iterations, for different input tokens, at the same time as we mid-train the other model parameters.

At each iteration $t$, after computing the hidden representation $h_t$ with the recursive block, the *router* outputs a halting values $w_t \in (0, 1)$ for each token. These values are accumulated across iterations:

$$H_t = \sum_{j=1}^{t} w_j. \tag{4}$$

For each input, the initial value of $H_t$ is zero. Computation for a token is stopped once $H_t \geq 1 - \epsilon$, with $\epsilon = 0.01$. However, early during training the router may output extremely small halting values, causing excessively many iterations. To avoid this, we cap the maximum number of iterations during training to $N_{max}$=10. During training we use the same hyperparameters as during fixed-depth ETD training. We do not provide auxiliary losses (e.g., intermediate losses after each iteration) nor we introduce any regularizers. Hyperparameters—including optimizer, learning rate, and scheduler—remain identical to fixed-depth ETD. The router is trained end-to-end jointly with the model.

**At test-time.** The test time regime is very similar to the training regime, except that once the model is trained we remove the cap on the number of iterations. The model determines on its own the number of iterations: after each iteration the router uses the output of the recursive block to predict the halting value for the iteration, and stops as soon as the cumulated halting values exceed $1 - \epsilon$: $\sum_{j=1}^{K} w_j > 1 - \epsilon$, where $K$ is the number of iterations.

Intuitively, until sufficient confidence is accumulated, the latent "thought" state continues to be updated. The final representation passed to latent deocder is the output of "thinking" block T after the final iteration. For easy tokens, the computation halts after few iterations, whereas difficult tokens may trigger more recursive reasoning steps. This design enables test-time computation scaling: the model dynamically allocates additional reasoning depth where beneficial

# E    RESULTS WITH ITERATIONS OVER DIFFERENT LAYERS

We fix the recursive "thinking" block size and vary its starting position from layer 2 to 12 in steps of 2, which is equivalent to increasing the size of the latent encoder $N_E$ from 1 to 11 in steps of 2.

Table 7: Results of the Encode–Think–Decode (ETD) method when varying the subset of layers in the recursive block. We report accuracy (Acc.) when increasing the size of the latent encoder $N_E$ from 1 to 11 in steps of 2, for each of six task categories (as defined in Sec. 3.2).

| Model | Params/ FLOPs | Factual Knowledge | Reading Comprehension | Commonsense Reasoning | Multi-Disciplinary Reasoning | BBH | Math. Reasoning |
|---|---|---|---|---|---|---|---|
| 1-4*2-11 | 16 / 20 | 37.92 | 55.53 | 64.82 | 44.99 | 31.23 | 25.6 |
| 3-4*2-9 | 16 / 20 | 37.43 | 56.93 | 65.87 | 46.9 | 29.80 | 27.31 |
| 5-4*2-7 | 16 / 20 | 37.58 | 56.51 | 66.86 | 49.03 | 32.21 | 26.8 |
| 7-4*2-5 | 16 / 20 | 38.1 | 56.14 | 66.74 | 48.41 | 31.67 | 28.27 |
| 9-4*2-3 | 16 / 20 | 37.7 | 53.46 | 65.52 | 45.71 | 31.05 | 27.35 |
| 11-4*2-1 | 16 / 20 | 37.67 | 54.79 | 64.45 | 45.18 | 30.93 | 24.63 |

# F    PERFORMANCE OF ETD ON EACH TASK

Table 2 reports the results of the Encode–Think–Decode (ETD) method with varying numbers of iterations over recursive "thinking" blocks, compared to the OLMo 2 1B baseline on 6 categories of tasks described in Sec. 3.2. In this section, we share the results for each individual tasks in Tables 8.

Table 8: Results of the Encode–Think–Decode (ETD) method with varying numbers of iterations over recursive "thinking" blocks, compared to the OLMo 2 1B baseline. Reported metrics include accuracy (Acc.) and relative improvement ($\Delta$, in %) with respect to the baseline. Parameter counts denote the number of distinct layers, while FLOPs correspond to the number of effective forward-pass layers.

| Model | Params/FLOPs | Natural Questions | | TriviaQA | | BoolQ | | OpenbookQA | | DROP | | HellaSwag | |
| | | Acc. | $\Delta$ | Acc. | $\Delta$ | Acc. | $\Delta$ | Acc. | $\Delta$ | Acc. | $\Delta$ | Acc. | $\Delta$ |
|---|---|---|---|---|---|---|---|---|---|---|---|---|---|
| Baseline | 16 / 16 | 20.98 | - | 54.12 | - | 72.0 | - | 52.8 | - | 31.761 | - | 69.7 | - |
| Ours (k=2) | 16 / 20 | 20.76 | (-1.01%) | 55.43 | (+2.43%) | 75.7 | +(5.14%) | 57.0 | (+7.95%) | 35.73 | (+12.5%) | 69.8 | (+0.14%) |
| Ours (k=3) | 16 / 24 | 19.97 | (-4.78%) | 55.13 | (+1.88%) | 76.0 | (+5.56%) | 57.4 | (+8.71%) | 34.82 | (+9.64%) | 69.6 | (-0.14%) |
| Ours (k=4) | 16 / 28 | 20.35 | (-2.99%) | 55.13 | (+1.8%)8 | 78.0 | (+8.33%) | 58.8 | (+11.36%) | 36.47 | (+14.81%) | 71.0 | (+1.87%) |
| Ours (k=5) | 16 / 32 | 20.53 | (-2.12%) | 55.93 | (+3.36%) | 76.4 | (+6.11%) | 61.0 | (+15.53%) | 38.086 | (+19.91%) | 70.4 | (+1%) |

| Model | Params/FLOPs | Social IQa | | WinoGrande | | CommonsenseQA | | ARC-Easy | | ARC-Challenge | | MMLU | |
| | | Acc. | $\Delta$ | Acc. | $\Delta$ | Acc. | $\Delta$ | Acc. | $\Delta$ | Acc. | $\Delta$ | Acc. | $\Delta$ |
|---|---|---|---|---|---|---|---|---|---|---|---|---|---|
| Baseline | 16 / 16 | 58.1 | - | 66.69 | - | 66.67 | - | 78.5 | - | 50.85 | - | 44.52 | - |
| Ours (k=2) | 16 / 20 | 62.9 | (+8.26%) | 66.85 | (+0.24%) | 67.40 | (+1.11%) | 78.4 | (-0.13%) | 58.36 | (+14.77%) | 47.59 | (+6.9%) |
| Ours (k=3) | 16 / 24 | 63.9 | (+9.98%) | 68.19 | (+2.25%) | 69.29 | (+3.93%) | 79.7 | (+1.53%) | 60.24 | (+18.46%) | 49.40 | (+10.96%) |
| Ours (k=4) | 16 / 28 | 65.0 | (+11.88%) | 68.51 | (+2.72%) | 68.14 | (+2.21%) | 79.8 | (+1.66%) | 62.03 | (+21.98%) | 49.84 | (+11.95%) |
| Ours (k=5) | 16 / 32 | 66.2 | (+13.94%) | 68.59 | (+2.84%) | (+68.47%) | (+2.7%) | 80.4 | (+2.42%) | 61.43 | (+20.81%) | 49.95 | (+12.19%) |

| Model | Params/FLOPs | MMLU Pro | | AGIEval English | | BBH | | GSM8K | | MATH | |
| | | Acc. | $\Delta$ | Acc. | $\Delta$ | Acc. | $\Delta$ | Acc. | $\Delta$ | Acc. | $\Delta$ |
|---|---|---|---|---|---|---|---|---|---|---|---|
| Baseline | 16 / 16 | 15.55 | - | 35.58 | - | 31.8 | - | 44.05 | - | 4.57 | - |
| Ours (k=2) | 16 / 20 | 17.53 | (12.72%) | 40.16 | (12.86%) | 31.67 | (-0.4%) | 51.10 | (+16.01%) | 5.45 | (19.22%) |
| Ours (k=3) | 16 / 24 | 18.13 | (+16.57%) | 40.27 | (+13.2%) | 32.62 | (+2.58%) | 54.36 | (+23.41%) | 6.22 | (+36.04%) |
| Ours (k=4) | 16 / 28 | 18.37 | (+18.12%) | 40.88 | (+14.89%) | 33.01 | (+3.82%) | 55.50 | (+25.99%) | 3.73 | (-18.28%) |
| Ours (k=5) | 16 / 32 | 19.07 | (+22.66%) | 42.07 | (+18.24%) | 33.49 | (+5.3%) | 56.56 | (+28.4%) | 4.33 | (-5.17%) |

# G    RESULTS WITH ITERATIONS OVER VARYING RECURSIVE BLOCK SIZE

We vary the block size by removing and adding layers symmetrically around the originally selected 7–4*2–5 configuration, keeping the recursive block centered in the same region of the model while changing its extent. We report the performance with sizes of recursive block of 2,4,6,8, and 12.

Table 9: Results of the Encode–Think–Decode (ETD) method when varying the number of layers in the recursive block. We report accuracy (Acc.) when the size of the recursive block $T$ is 2, 4,6,8, and 12, for each of six task categories (as defined in Sec. 3.2).

| Model | Params/ FLOPs | Factual Knowledge | Reading Comprehension | Commonsense Reasoning | Multi-Disciplinary Reasoning | BBH | Math. Reasoning |
|---|---|---|---|---|---|---|---|
| 8-2*2-6 | 16 / 18 | 37.99 | 55.23 | 65.88 | 47.00 | 30.98 | 26.63 |
| 7-4*2-5 | 16 / 20 | 38.10 | 56.14 | 66.74 | 48.41 | 31.67 | 28.27 |
| 6-6*2-4 | 16 / 22 | 38.37 | 57.43 | 67.01 | 49.09 | 31.81 | 29.04 |
| 5-8*2-3 | 16 / 24 | 37.00 | 57.67 | 67.73 | 49.54 | 33.71 | 29.35 |
| 2-12*2-2 | 16 / 28 | 37.70 | 56.44 | 67.73 | 47.58 | 32.30 | 29.27 |

# H    RESULTS ON OLMO-2 7B MODEL

# I    TRAINING COMPUTE OVERVIEW

We run all our experiments on a node with 8 A100 80GB GPUs. Table **??** we report training time of experiments presented in Table 2.

Table 10: Results with larger model and same FLOPs

| Model | Params/FLOPs | Factual Knowledge | Reading Comprehension | Commonsense Reasoning | Multi-Disciplinary Reasoning | BBH | Math. Reasoning |
|---|---|---|---|---|---|---|---|
| OLMo 2 7B | 32 / 32 | 56.63 | 74.68 | 76.73 | 62.9 | 48.18 | 41.63 |
| 16-10*2-6 | 32 / 40 | 56.89 | 75.05 | 76.82 | 62.95 | 49.77 | 42.64 |

Table 11: Compute cost of experiments (GPU hours per full training run).

| Model | GPUs | Hours / run | GPU hours |
|---|---|---|---|
| OLMo2 (k=1) | $8 \times$ A100 | ~116 | ~928 |
| ETD (k=2) | $8 \times$ A100 | ~137 | ~1,096 |
| ETD (k=3) | $8 \times$ A100 | ~170 | ~1,360 |
| ETD (k=4) | $8 \times$ A100 | ~195 | ~1,560 |
| ETD (k=5) | $8 \times$ A100 | ~220 | ~1,760 |

## J  FUTURE WORK

Future work spans several directions. Extending ETD to multimodal models could establish recursive latent reasoning as a general principle of representation learning across domains. Designing more efficient training strategies, together with refining adaptive depth mechanisms, may yield better compute–performance trade-offs. Assessing the impact of ETD on instruct models will require integration at the post-training stage, which we leave for future investigation. Last but not least, conducting interpretability studies could clarify how recursive latent reasoning interacts with model circuits and representations, offering deeper insights into the structure of reasoning in LLMs.

## K  USAGE OF LARGE LANGUAGE MODELS

In preparing this manuscript, we used large language models (LLMs) solely as writing assistants, to improve grammar, style, and clarity. The authors retain full responsibility for the content and any remaining errors.

