# OpenReview forum: "Encode, Think, Decode: Scaling test-time reasoning with recursive latent thoughts"
_ICLR.cc/2026/Conference — Submitted to ICLR 2026_

### Official Review · Reviewer_8tY2 · 2025-10-14

**Soundness:** 1
**Presentation:** 2
**Contribution:** 2
**Rating:** 2
**Confidence:** 4

**Summary:**

This paper introduces Encode-Think-Decode (ETD), a method for converting Olmo-2-1b into a recursive reasoning model. The paper uses angular distance between layer outputs and the kneedle algorithm (for finding distinct changes) to define how to partition the model into Encoder, Recursive Block and Decoder. Later this decision is empirically validated, showing that for this model benchmarks tend to be higher in the region the algorithm finds (Fig. 3).

The paper shows gains over a selection of benchmarks from the OLMOes suite, showing multiple different recursive depths. The authors compare to the baseline (non-recursive) Olmo-2 model, finding that they often increase accuracy by between 2 and 5% in absolute terms on average but do use more FLOPs during training and inference.

The authors also baseline to different partitions of layers into the Encoder, Recurrent Block, Decoder. Finding a 2-12\*2-2 model is very similarly matched to their 7-4\*4-7 model and FLOPs equivalent. Finally, the authors train a linear probe for the latent space of the recurrent block which is used for early exiting, finding a trade off between efficiency and cost in most cases.

**Strengths:**

1. Analyses post training models to be recursive, an approach which may reduce cost of future research.
2. Increases benchmark performance on average for Olmo-2-1b after mid-training, when using more FLOPs.
3. Uses an interpretable approach to selecting layers for the recurrent block which is empirically verified.

**Weaknesses:**

### Claims:
- Some of the papers claims are overstated:
    - The use of relative improvement feels abused here. For example: claiming +36% relative improvement on MATH in the abstract and only achieving a 1.65% absolute improvement feels misleading to the reader.
    - The claim of +28% on GSM8K and +36% on MATH are regularly stated together. However, these are not achievable together with the same k value, this is never clearly stated.
    - The paper claims “substantial” gains on 17 benchmarks, However, these are small in absolute terms.
- In Table 4, when compared to baselines the margin of results becomes extremely small. What are the error bounds on these experiments as the 2-12\*2-2 model is very similarly matched to their 7-4\*4-7 model? Did the authors explore this further, perhaps training a 2-12\*4-2 model is even better? This also increases my concerns that it is the FLOPs used for computation and less so the proposed method driving performance increases.

### Scope:
- The paper only considers one model Olmo-2-1b.
- The is no FLOPs matched non-recursive baseline training run. For example, this increase in accuracy may be due to more computation during training and inference.
    - The paper states: “Our goal in applying a recursive approach, conversely, is to boost reasoning capabilities by efficiently scaling inference-time computation.” However, I do not think this is a fair reason to eschew a FLOPs matched baseline.
- On line 199, the authors claim access to the training data is required to run these types of experiments, however I think they can be conducted without access to training data as long as the same training data is used for all models being analysed.

### Relation to prior work:
- A lot of the citations used when trying to distinguish from prior work by highlighting the approach of using angular distance to locate layers are for models trained from scratch hence, I think these aren’t quite the right citations (e.g. line 149).
- Line 185: “Prior works on recursive-depth models typically rely on simplified training setups.” Geiping et al. and Bae et al. (2025) train in standard set ups for long periods also.
- In the “Key differences to prior work” many points are highlighted such as LoRA Adapters, Regularisation and Input Injections. However, there is a lot of nuance missed here, for example Geiping et al. train for a large number of iterations but this also allows extrapolation in terms of k. Moreover, Aleksandrov et al. find, like much other prior work, input injection is useful. Without baselining against these methods to show the new proposal is superior, I find this weak evidence of novelty.

**Questions:**

1. Why is Figure 1 taken over C4 and not the training set? Or some other dataset meant to match the reasoning distribution the authors are targeting?
2. Is the baseline “Olmo 2 (k=1)” model trained by the authors? I worry here about the authors training set up differing from the Olmo suite in minor ways leading to the small changed in accuracy we see.
3. I have a large number of questions about Section 5:
    1. What is the training data?
    2. Is the whole model being trained or just the router?
    3. Do the authors have any reasoning for the increase in DROP and OpenbookQA?
    4. What value of K is the model trained with? If it can extrapolate to maximum k=10, one would assume 10 which is not shown as a result elsewhere in the paper.
    5. What objective is used when training the router?

During rebuttal, I would be most interesting in clarifications on the baselines accuracy and training, more baselines being considered and more architectures being considered.

---

> ### Author Response · Authors · 2025-11-21
> **Author's rebuttal (part 1)**
>
> We first would like to thank the reviewers for their feedbacks and comments. Please find below our responses addressing the weaknesses.
>
> ## Reply to weakness related to "Claims"
>
> ### 1. Re: Usage of relative improvements
>
> Our goal in reporting relative improvements was not to overstate the results but to normalize performance gains across tasks with very different starting accuracies.
> For example, ETD (k = 3) yields similar absolute gains on MATH (+1.65%) and TriviaQA (+1.01%), but the baselines differ substantially (4.57% vs. 54.12%). In such cases, absolute improvements alone do not reflect the magnitude of progress, since the same absolute gain can correspond to very different fractional effects depending on the baseline.
> For this reason, we report both absolute performance and relative improvements for all 17 evaluation tasks directly in the tables, so that readers can interpret the results using whichever measure they find more informative.
>
> ### 2. Re: On claim of +28% on GSM8K and +36% on MATH stated together
>
> Thank you for pointing this out — we understand how the original wording could give the impression that the +28% and +36% improvements on GSM8K and MATH are achieved simultaneously with the same k value. *Our intention was instead to summarize the maximum relative gains that ETD produces across tasks.*
>
> To clarify the exact results: both MATH (+36%) and GSM8K (+23%) are obtained at k = 3, while the +28% GSM8K improvement corresponds to k = 5. Although the optimal k varies slightly across benchmarks, the trend remains consistent — ETD substantially improves reasoning performance across tasks.
>
> To avoid ambiguity, we have updated the abstract (blue-highlighted text) to explicitly state that the results in the abstract correspond to the maximum gains with our method.
>
> ### 3. Re: On gains on 17 benchamarks
>
> Thank you for raising this point. The 17-benchmark suite was selected to evaluate ETD across tasks with very different reasoning requirements. As shown in Section 4, ETD yields large performance increases on reasoning-intensive tasks such as MATH and GSM8K, whereas tasks that rely mostly on factual recall or surface-level patterns (e.g., BoolQ, TriviaQA) show much smaller gains. This pattern is consistent with our hypothesis that ETD amplifies latent reasoning rather than general knowledge memorization.
>
> In this context, our use of the term “substantial” gains refers specifically to the substantial improvements on the reasoning-focused tasks, which are the primary target of ETD. We agree that this distinction was not emphasized clearly enough in the original claim.
>
> To avoid any ambiguity, we revised the statement in the paper to highlight that ETD achieves on average notable improvements on 17 benchmarks .
>
> ### 4. Re: On 2-12x2-2 baseline
>
> The results in Table 4 results show small but consistent gains across all task groups. Results for 2-12x2-2 and 7-4x4-5 models compare them in FLOP-equivalent setup. However, we want to highlight that even the 7-4*3-5 configuration, with less FLOPs, shows better performance than 2-12x2-2 on reasoning-intersive tasks.
>
> We understand the reviewer’s concern that the improvements might be driven by additional FLOPs rather than the ETD mechanism. To address this, Figure 2 (and Table 6 for tabular presentation) reports ablations where FLOPs are held constant while only the choice of layers included in the recursive block is varied. These results clearly show that, under the same computation budget, the position and composition of the recursive block are the main drivers of performance on reasoning tasks.

---

> ### Author Response · Authors · 2025-11-21
> **Author's rebuttal (part 2)**
>
> ## Reply to weakness related to "Scope"
>
> ### 1. Re: on using OLMo-2 1B model
>
> Thank you for the thoughtful suggestion — we agree that evaluating ETD on additional model families is an interesting direction. As the reviewer Qf7A also asked about our choice of experimental setup, we address this point in the general response to all reviewers.
>
> ### 2. Re: FLOPs matched non-recursive baseline
>
> Thank you for the suggestion. We agree that a FLOPs-matched baseline is a useful comparison to further separate the effect of recurrence from simply increasing computation. Our original scope, however, was to isolate the effect of recurrence without changing model capacity, which is why we kept the number of parameters fixed. Matching FLOPs instead would require increasing the parameter count and introduces a different confound—making it difficult to attribute performance differences specifically to recurrence.
>
> Within this scope, Figure 2 already shows that performance is not driven purely by additional compute: when holding FLOPs constant and varying only where the recursive block is placed, performance changes substantially, indicating that the allocation of computation is more important than raw FLOP count.
>
> That said, we appreciate the reviewer’s interest in the FLOPs-matched comparison and have now run this additional experiment. We contrast:
> - 7-4×2-5 (ETD: recurrence over a 4-layer block, looped twice), and
> - 7-8×1-5 (non-recurrent: doubling depth to match FLOPs, initialized middle 8 layer by stacking the same 4-layer block).
> Stacking the same 4-layer block to mimic the two iterations while keeping the FLOPs constant. Both configurations perform identically before mid-training. Both systems use identical data, hyperparameters, and checkpoints. Results are as follows:
>
> | Model       | Params / FLOPs | Factual Knowledge | Reading Comprehension | Commonsense Reasoning | Multi-Disciplinary Reasoning | BBH  | Math. Reasoning |
> |-------------|----------------|-------------------|------------------------|------------------------|------------------------------|------|------------------|
> | OLMo-2      | 16 / 16        | 37.55             | 52.19                  | 65.29                  | 45.00                        | 31.80 | 24.31           |
> | 7-8-5       | 20 / 20        | 31.78             | 52.03                  | 62.45                  | 44.42                        | 30.21 | 21.68           |
> | **7-4×2-5** | 16 / 28        | **38.10**         | **56.14**              | **66.74**              | **48.41**                    | **31.67** | **28.27** |
>
> Key property of ETD is that it leverages the pretrained model efficiently.Results in table above show that the larger iso-FLOPs model underperforms both the original non-recurrent baseline and the ETD (k=2) model, indicating that simply increasing effective depth does not reproduce the improvements achieved by ETD.
> We added a new subsection in Section 4 with results of new ablations (highlighted in blue in the revised version).
>
>
> ## 3. Re: On importance of access to training data for controlled experiments
>
> It is possible to conduct such experiments without access to the original pre-training data as long as all models are trained on the same additional data. Our concern is that this setup may introduce bias if the additional data overlaps differently with the models’ pre-training corpora: a model that has not seen that data before would benefit more than a model that already has, making it difficult to attribute performance changes to recurrence rather than to differences in data exposure.
>
> This effect has been observed empirically. Bae et al. (ICLR’2025) fine-tuned different models on the same publicly available dataset and found that the non-recurrent baseline lost performance, making it challenging to isolate whether subsequent changes were due to architectural changes or due to how suitable the additional dataset was for the model.
>
>
> [1] “Relaxed Recursive Transformers: Effective Parameter Sharing with Layer-wise LoRA”

---

> ### Author Response · Authors · 2025-11-21
> **Author's rebuttal (part 3)**
>
> ## Reply to weakness related to "Prior Works"
>
> ### 1. Re: On using citations
>
> ETD is original in adapting an existing pretrained model into a recurrent one, rather than retraining from scratch. We cite prior recurrent approaches not to imply that they operate in the same setting as ETD, but to position ETD within the broader landscape of recurrent architectures. To avoid confusion, we will revise the related-work section to more clearly distinguish between approaches that train recurrent models from scratch and methods like ETD that adapt pretrained models.
>
> ### 2. Re: On concurrent work's training setup
>
> Thank you for pointing this out. Current wording can be interpreted more strongly than intended. Our statement that “prior works on recursive-depth models typically rely on simplified training setups” refers not to the scale or duration of training, but to the degree to which the full pre-training pipeline is exercised. In works such as Geiping et al. and Bae et al., recursive architectures are evaluated without conducting the extensive hyperparameter search, data-mixture tuning, and training-recipe optimization that are commonly applied when developing production-grade LLMs.
>
> By contrast, ETD is applied to an already fully optimized open-source model trained with a state-of-the-art architecture, data mixture, and training recipe. Our intent was to highlight that ETD improves models that have already undergone a highly optimized pre-training pipeline, rather than requiring a simplified or separate training pipeline for recurrence.
>
> To prevent any misunderstanding, we revised the text (blue colored) to make this distinction clearer.
>
>
> ### 3. Clarification on Key Differences from Prior Work
>
> Thank you for raising this point. We agree that prior work such as Geiping et al. and Aleksandrov et al. introduces valuable techniques — e.g., input injection, regularization strategies, and layer-specific adapters — and that these methods can benefit recursive-depth architectures when the model is trained from scratch. However, these mechanisms are not directly applicable to a pretrained model.
>
> Our focus in this paper is complementary rather than overlapping: ETD does not require any architectural modification, auxiliary loss, or retraining from scratch, and can be applied to an existing model without altering its parameters, training data, or training recipe. In this sense, the novelty of ETD lies not in proposing a more complex recurrent architecture, but in demonstrating that a simple recurrence mechanism applied to a pretrained LLM can improve reasoning ability without any additional components or pre-training.
>
> Comparing against all alternative training-from-scratch recurrent architectures is outside the scope of this work, as they optimize a different objective. ETD targets a distinct and practical use case: enabling recurrence for models that have already been trained.

---

> ### Author Response · Authors · 2025-11-21
> **Author's rebuttal (part 4)**
>
> ## Replies to Questions
>
> ### 1. Re: Why is Figure 1 taken over C4 and not the training set?
>
> We follow the protocol of Gromov et al. (2024, ICLR) for identifying structural transitions in pretrained Transformers, where layer-to-layer dynamics are computed on the C4 validation set. Our goal is to estimate the encoder/recursive block/decoder segmentation in a task-agnostic and generalizable way rather than optimizing it for a particular evaluation domain. Using a reasoning-heavy dataset (e.g., mathematics) would risk biasing the segmentation toward those domains. C4 provides a neutral, broad-coverage corpus that is widely used for studying representational structure in pretrained LLMs without introducing target-task bias.
>
> ### 2. Re: Is the baseline “Olmo 2 (k=1)” model trained by the authors?
>
> Yes, the “OLMo-2 (k = 1)” baseline is trained by us. The official OLMo-2 1B Base model is released as a “soup” of three mid-training runs, so to ensure a fair comparison we follow the official training recipe starting from the same publicly released stage-1 checkpoint and run a single mid-training run using the official codebase, hyperparameters, data pipeline, and learning-rate schedule. This avoids any confounders that could arise from comparing a merged model to a single-seed ETD run.
>
> For ETD, we use exactly the same training setup and the same seed as the baseline. The only difference between the two runs is the use of recurrence during mid-training. Therefore, the small accuracy changes reflect the effect of ETD rather than differences in training setup.
>
> ### 3. Re: Missing details in Section 5
>
> We thank the reviewer for raising these questions — we agree that the submitted draft did not sufficiently clarify the adaptive-depth setup. We have now expanded Section 5.1 with the necessary details.
>
> The adaptive-depth ETD training pipeline is identical to fixed-depth ETD: we load the official OLMo-2 1B stage-1 checkpoint, initialize the router randomly, and continue mid-training on the same dataset using the same training recipe. The entire model, including the router, is trained end-to-end; we do not freeze any components.
>
> No auxiliary losses, reinforcement learning, depth labels, or handcrafted supervision signals are introduced. The router is trained purely through the standard language-modeling loss, backpropagated through all unrolled iterations of the recursive block. During training, the maximum possible number of iterations is 10.
>
> To ensure clarity on training details of adaptive-depth ETD, we also added the clarifications in Section 5.1 (highlighted in blue in the updated manuscript).
>
> The results of adaptive-depth in DROP and OpenbookQA are indeed interesting. We conjecture that reading comprehension tasks have natural stopping points where sufficient information has been extracted and integrated. The ACT-based halting mechanism appears to be particularly effective at detecting these points, leading to more efficient and accurate processing compared to predetermined iteration counts. More investigations are necessary to verify this conjecture

---

> ### Comment · Reviewer_8tY2 · 2025-11-25
> **Rebuttal Response**
>
> Thank you for your rebuttal, I have limited my response to where I have further questions but have read all points and acknowledge the effort put in by the authors.
>
> # Claims:
> 4. While I find the point about the performance increase with less FLOPs compelling, is this a significant finding (2 standard errors) as evaluations are inherently noisy?
>
> ## New question:
> 5. If I understand the rebuttal update correctly, my concern is resolved in Figure 3. I think if I could see the new Figure 3 with error bars this would put most of my method concerns to bed one way or another. For example the green box from what I understand this is *not* ETD, as it doesn't use kneedle, but is as competitive as ETD in almost all cases. Same could be said to a lesser extent of the yellow triangle and purple diamond, but if the gap to ETD is more than 2 standard errors/deviations, to me this is the solid proof of the method for Olmo.
>
> # Scope
> 1. Thank you for the additional experiments (would be nice to see the kneedle result in the new draft for this). However, it is very difficult to tell if these findings are more than noise without standard error/deviation as a 0.84% increase in gsm8k is quite small.
>
> 2. I think the FLOPs matched baseline here would be in terms of data as the authors say further down in the rebuttal “the novelty of ETD lies not in proposing a more complex recurrent architecture, but in demonstrating that a simple recurrence mechanism applied to a pretrained LLM can improve reasoning ability without any additional components or pre-training.” Hence, the null hypothesis to me here is training the regular model, perhaps epoching (https://arxiv.org/pdf/2305.16264) the data without adding new parameters.
> I understand it may be late to run new training experiments but the authors could also take an earlier checkpoint from the recursive model approximately flops matched to the non-recurrent baseline.
>
> 3. There is lots of new synthetic data released on Hugging Face since the release of Olmo-2, these datasets may work and are expansive.
>
> # Prior work
> 3. This section of the draft is still worded as though the authors try to claim additional novelty. I think this section should be reworded in line with the response above.
>
> # Questions
> 3. “During training, the maximum possible number of iterations is 10.” Why, as k=10 is not shown elsewhere in the paper? Does this mean more FLOPs are used when training compared to the fixed recursion baselines?
>
> # New Concern
> 1. Page limit is exceeded.
>
> ---
>
>
> I agree with H7YH on the novelty of the ETD as it stands.
>
> I would be interested to hear the thoughts of other reviewers on the use of relative accuracy as to me it seems like an over claim.

---

> > ### Author Response · Authors · 2025-12-02
> > **Author's Response (part 1)**
> >
> > We thank the reviewer for the feedback and are glad that most of the raised weaknesses and questions are now resolved. Below we address the remaining follow-up points.
> >
> > ## On claims:
> > We are glad that our rebuttal update resolved your previous concerns. We perform the mid-training on the 1B parameter model and we believe that it is not standard practice to report error bars on experiments of this scale neither for recursive-depth [1, 2, etc] nor for standard  models [3, etc].  The table below reports the training time of experiments presented in Table 2. We ran all experiments on a node with 8 A100 80GB GPUs.
> >
> > | Model       | GPUs        | Hours / run | GPU hours |
> > |-------------|-------------|-------------|-----------|
> > | OLMo2 (k=1) | 8 × A100     | ~116        | ~928      |
> > | ETD (k=2)   | 8 × A100     | ~137        | ~1,096    |
> > | ETD (k=3)   | 8 × A100     | ~170        | ~1,360    |
> > | ETD (k=4)   | 8 × A100     | ~195        | ~1,560    |
> > | ETD (k=5)   | 8 × A100     | ~220        | ~1,760    |
> >
> > To answer your request, in addition to our original experiments we trained the 7-4*2-5 model with two more random seeds. We can clearly see that the results are very consistent, with very small standard deviations. While ideally statistical significance would be determined on numerous seeds, these results are evidence that differences between ETD and the baselines cannot be ascribed to noise.
> >
> > | Model          | Params / FLOPs | **Factual Knowledge** Avg. | Std.  | **Reading Comprehension** Avg. | Std.  | **Commonsense Reasoning** Avg. | Std.  | **Multi-Disciplinary Reasoning** Avg. | Std. | **BBH** Avg. | Std.  | **Math. Reasoning** Avg. | Std.  |
> > |---------------|----------------|---------------------------|-------|-------------------------------|-------|--------------------------------|-------|----------------------------------------|-------|-------------|-------|---------------------------|-------|
> > | 7-4x5-2  | 16 / 20        | 37.96                    | 0.0075     | 56.23                        | 0.0088     | 66.78                        | 0.0065     | 48.52                                    | 0.0025     | 31.69        | 0.0167     | 27.59                    | 0.0228     |
> >
> >
> > [1]  “Scaling up Test-Time Compute with Latent Reasoning:  A Recurrent Depth Approach” (https://arxiv.org/abs/2502.05171)
> >
> > [2] “Relaxed Recursive Transformers: Effective Parameter Sharing with Layer-wise LoRA” (https://arxiv.org/abs/2410.20672)
> >
> > [3] “2 OLMo 2 Furious” (https://arxiv.org/abs/2501.00656)

---

> > ### Author Response · Authors · 2025-12-02
> > **Author's Response (part 2)**
> >
> > ## On Scope
> >
> > ### Re: 1. On 7B results:
> > The impact of ETD on the 7B baseline, OLMo-2 7B Base model (i.e. after mid-training), in absolute gains is less significant than on the 1B baseline, OLMo-2 1B Base model, and we appreciate the reviewer’s caution.
> >
> > Due to extreme computational costs we cannot run mid-training on 7B models several times to estimate the noise in the measure: this is an unfortunate reality that all research on LLMs of this size is faced with.
> >
> > We want to bring two points that may help contextualize the results on 7B baseline:
> > 1. Importantly, **the largest gains at 7B again appear on mathematical reasoning** — the category ETD is designed to benefit by looping over reasoning-critical layers — validating that the ETD mechanism works across scales.
> > 2. Because both 1B and 7B were trained on the same 50B tokens during mid-training, the 1B model is exposed to more data per parameter, which naturally amplifies relative improvements in that regime.
> >
> >
> > ### Re: 2. On FLOPs matched baseline on data
> > In Section 4 we define FLOPs in our paper in terms of forward passes through layers. We use this definition though the paper and rebuttal discussions.
> >
> > We assume  that the reviewer implied the FLOPs ~= (effective parameter count) x (number of training tokens). As you have mentioned we cannot run new training experiments, but upon your request we took an earlier checkpoint from the recursive model. The recursive model has 20 effective layers while non-recursive has 16 layers. The full mid-training consists of 23’852 steps, so to match the training compute we take the checkpoint after 16/20 * 23’852 ~= 19000 steps
> >
> > The table with results below show that ETD approach outperforms non-recursive models while being trained on 20% less data
> >
> > | Model        | Params / FLOPs | Training Steps | **Factual Knowledge** Acc. | Δ (%) | **Reading Comprehension** Acc. | Δ (%) | **Commonsense Reasoning** Acc. | Δ (%) | **Multi-Disciplinary Reasoning** Acc. | Δ (%) | **BBH** Acc. | Δ (%) | **Math. Reasoning** Acc. | Δ (%) |
> > |-------------|----------------|----------------|-----------------------------|-------|--------------------------------|-------|----------------------------------|-------|----------------------------------------|-------|-------------|-------|---------------------------|-------|
> > | OLMo 2 (k=1) | 16 / 16        | 23852          | 37.55 | – | 52.19 | – | 65.29 | – | 45 | – | 31.8 | – | 24.31 | – |
> > | **ETD (k=2)** | 16 / 20       | 19000          | 37.96 | (+1.09%) | 56.62 | (+8.49%) | 66.79 | (+2.3%) | 47.43 | (+5.4%) | 31.36 | (−1.39%) | 27.25 | (+12.09%) |
> > | **ETD (k=3)** | 16 / 24       | 19000          | 36.81 | (−1.97%) | 55.62 | (+6.57%) | 68.25 | (+4.53%) | 48.92 | (+8.71%) | 32.02 | (+0.68%) | 29.01 | (+19.33%) |
> > | **ETD (k=4)** | 16 / 28       | 19000          | 37.00 | (−1.46%) | 57.10 | (+9.41%) | 67.96 | (+4.09%) | 49.59 | (+10.2%) | 32.83 | (+3.24%) | 29.71 | (+22.21%) |
> > | **ETD (k=5)** | 16 / 32       | 19000          | 38.12 | (+1.52%) | 57.71 | (+10.58%) | 68.21 | (+4.47%) | 50.16 | (+11.47%) | 32.37 | (+1.78%) | **30.18** | (+24.15%) |
> >
> > ### Re: 3. Fine-tuning on new synthetic data
> > Thank you for this suggestion. We covered the motivation for our experimental setup in the general reply to all reviewers. We will experiment with synthetic data in future work.

---

> > ### Author Response · Authors · 2025-12-02
> > **Author's Response (part 3)**
> >
> > ### On Prior Work
> >
> > It is difficult to address this feedback without specifics of “additional novelty”. In the updated version of the paper we reflected all the feedback from all reviewers. We hope the updated version of the paper will resolve the reviewer's concerns.
> >
> > ### On Questions about ACT training:
> >
> > Since the model learns to decide when to continue/stop looping, in the beginning of training the router can allow too many iterations. Therefore, we cap the maximum number of iterations during training to 10.
> >
> > We have added a new section in the Appendix with Details on Adaptive-depth ETD training.
> >
> > ### On Page Limit concern:
> > We resolved the Page Limit issue.

---

### Official Review · Reviewer_cBrU · 2025-10-23

**Soundness:** 2
**Presentation:** 3
**Contribution:** 3
**Rating:** 4
**Confidence:** 3

**Summary:**

The paper introduces a simple yet effective method for converting existing models to recurrent models by training the model to iterate over a small subset of the "reasoning"-relevant layers during mid-training. The paper shows the benefits of more "reasoning" benchmarks such as GSM8K and MATH.

**Strengths:**

- simple but effective method for increasing the performance of the model on GSM8k and MATH
- introduce a mechanism to adaptively determine the number of iterations for each input

**Weaknesses:**

- results only on a single model. This is important for this study because the layers found to iterate over may be specific to the evaluations themselves.
- How do methods like these compare to latent approaches like COCONUT [1] or similar? Although no baseline is necessarily directly comparable, for a better understanding, it would be useful to include another method as a comparison point.

[1] https://arxiv.org/abs/2412.06769

**Questions:**

- How were the evaluations grouped into reasoning/non-reasoning benchmarks?
- How can we confirm that the current results on layer choosing is not an artifact of the OLMo models?
- Are different model sizes explored in the paper?

---

> ### Author Response · Authors · 2025-11-21
> **Author's rebuttal (part 1)**
>
> ## Reply to Weakness 1:
>
> Thank you for highlighting this important point. As Reviewer Qf7A raised a closely related concern regarding the experimental setup, we address the broader rationale in the general response to all reviewers.
>
> Regarding whether the learned layer partition might be evaluation-specific: we examined this possibility carefully. Our evaluations span 17 diverse benchmarks. If the partition were overly tuned to a specific evaluation type, we would expect regressions on non-reasoning tasks — yet we do not observe degradation on any of the 17 benchmarks.
>
> This said, Figure 2 does suggest that the discovered partition reflects functional roles of layers:
>
> – For reasoning-intensive tasks (e.g., mathematical reasoning), recursively iterating the mid “thinking” block yields the strongest gains.
>
> – For reading-comprehension-style tasks, iterating over a different subset of layers is most effective.
>
> While a fuller causal understanding of why these roles differ remains an open research direction, our results indicate that (i) the approach generalizes across a wide range of tasks, and (ii) improvements in reasoning do not come at the cost of performance elsewhere.

---

> ### Author Response · Authors · 2025-11-21
> **Author's rebuttal (part 2)**
>
> ## Reply to weakness 2
>
> Thank you for raising this point — we agree that situating ETD in the context of latent-reasoning approaches such as COCONUT is helpful for understanding its contribution.
>
> COCONUT alternates between a language mode (standard autoregressive decoding) and a latent mode in which the model feeds the hidden state directly as the next input embedding. This requires a multi-stage training curriculum to teach the model to operate effectively in both modes. In contrast, ETD preserves the original forward pass and does not introduce new latent execution modes or additional training stages; instead, it redistributes computation by recursively executing a subset of existing layers.
>
> The closest analogue to COCONUT in our study is the E = D = 0 configuration in Table 4, where the recursive loop applies to the entire depth of the model (i.e., repeated latent transformation without intermittent decoding). This variant underperforms ETD under the same FLOPs budget, suggesting that isolating a dedicated “thinking” block of layers is key. We also note that the original COCONUT work does not improve over its baseline on GSM8K, further indicating that full-depth latent looping is not effective for mathematical reasoning.
>
> To clarify the relationship between the two lines of work, we added a more detailed discussion in the related-work section summarizing the methodological differences (text in blue in “Key Differences from Prior Work”, Section 5). We believe this framing helps position ETD within the broader landscape of methods focusing on reasoning in continuous latent space.

---

> ### Author Response · Authors · 2025-11-21
> **Author's rebuttal (part 3)**
>
> ## Reply to Question 1:
>
> The grouping follows criteria discussed in Appendix B, where we categorize benchmarks along a spectrum from non-reasoning, to reasoning + information-retrieval, to predominantly reasoning-centric tasks. Each dataset’s placement is justified based on the level of implicit reasoning, multistep inference, and reliance on background knowledge required to answer a query.
>
> This categorization aligns with prior literature, including the taxonomy used by Saunshi et al. (ICLR 2025) [2], and we extend it to cover a broader set of benchmarks. Importantly, our conclusions do not depend on the exact grouping choice: per-task results are reported individually (Table 5), and the gains from ETD are observable at the dataset level even before aggregation into categories.
>
> [2] https://arxiv.org/abs/2502.17416
>
>
> ## Reply to Question 2:
>
> Thank you for raising this concern. ETD does not rely on any architectural features unique to OLMo models: the method requires only residual-stream activations and a standard transformer block that can be executed recursively, which are properties shared by most decoder-only autoregressive LLMs. For this reason we expect the approach to be applicable beyond OLMo, and we explicitly design ETD to avoid model-specific components or retraining.
> We agree that validating ETD on additional architectures is an important direction for future work. In the general reply to all reviewers we explain why, for this initial study, we focused on OLMo (open weights, complete training stack access, and controlled experimentation across scales). The goal of the present paper is to establish the method and its mechanisms, and we view extension to other model families as a natural follow-up rather than evidence that the current results are tied to OLMo-specific artifacts.
>
>
> ## Reply to Question 3:
>
> Thank you for raising this question. We have extended our experiments to OLMo-2 7B, to the best of our knowledge, we present the largest model trained with a recurrent architecture.  We dive into details in the reply to all reviewers.
>
> We also added a new subsection in Section 4 with results of new ablations (highlighted in blue in the revised version).

---

> > ### Comment · Reviewer_cBrU · 2025-11-25
> >
> > I thank the authors for their time running experiments and for their response.
> >
> > > Weakness 1 & Q3
> >
> > I do not think the results presented on 7B are significant (i.e, due to noise, hyperparameter setting, etc). These results seem marginal and are unclear whether they show a real improvement. Due to this reason, I will maintain my current score.
> >
> > > Weakness 2
> >
> > Thank you for further updating the paper. I do think adding a line in the results section that explicitly states something along the lines of "in the setting, E = D = 0 configuration in Table 4, is the closest analogue to COCONUT."
> >
> > I would like to mention that there are follow-ups to COCONUT that might be good comparison points as well. The idea of this experiment is not a direct comparison but a general comparison across different types of increasing test-time compute. I actually don't care what the results look like for this baseline. It is just important to compare the two types of continuous test-time reasoning. It would actually add a contribution to the paper for me.
> >
> > > Q1 & Q2
> >
> > Thank you for your response. I do not have any follow-ups on these questions.

---

> > > ### Author Response · Authors · 2025-11-27
> > > **Author's reply to Official Comment**
> > >
> > > We thank the reviewer for the constructive feedback and are glad that all earlier questions are now resolved. Below we address the two remaining follow-up points.
> > >
> > > ## On the 7B results.
> > > The impact of ETD on the 7B baseline, OLMo-2 7B Base model (i.e. after mid-training), in absolute gains is less significant than on the 1B baseline, OLMo-2 1B Base model, and we appreciate the reviewer’s caution.
> > >
> > > Due to extreme computational costs we cannot run mid-training on 7B models several times to estimate the noise in the measure: this is an unfortunate reality that all research on LLMs of this size is faced with.
> > >
> > > We want to bring two points that may help contextualize the results on 7B baseline:
> > > 1. Importantly, *the largest gains at 7B again appear on mathematical reasoning* — the category ETD is designed to benefit by looping over reasoning-critical layers — validating that the ETD mechanism works across scales.
> > > 2. Because both 1B and 7B were trained on the same 50B tokens during mid-training, the 1B model is exposed to more data per parameter, which naturally amplifies relative improvements in that regime.
> > >
> > >
> > >
> > > ## On the comparison with latent-iteration approaches such as COCONUT
> > >
> > > As suggested, we added an explicit statement clarifying that the “E=D=0 configuration is the closest analogue to COCONUT-style continuous latent iteration” in the Section 4..2 (highlighted in blue).
> > >
> > > We also expanded our discussion on the continuous test-time reasoning in our related work by adding the follow-up analysis of COCONUT approach [1,2]. If there is any specific related work that would be relevant to include please let us know.
> > >
> > > We hope that these additions resolve the remaining concerns and help clarify both (i) the mechanism underlying ETD across scales and (ii) how ETD fits within the space of continuous test-time reasoning approaches.
> > >
> > > [1] "Reasoning by Superposition: A Theoretical Perspective on Chain of Continuous Thought", Zhu et al. 2025 (https://arxiv.org/abs/2505.12514)
> > >
> > > [2] "Emergence of Superposition: Unveiling the Training Dynamics of Chain of Continuous Thought", Zhu et al. 2025 (https://arxiv.org/abs/2509.23365)

---

### Official Review · Reviewer_Qf7A · 2025-10-25

**Soundness:** 3
**Presentation:** 3
**Contribution:** 3
**Rating:** 6
**Confidence:** 3

**Summary:**

This paper follows the recursive transformer idea and proposes Encode–Think–Decode, which decomposes a pretrained LLM into three functional components, latent encoder, recursive “thinking” block, and latent decoder, based on layerwise representational dynamics. Using the mean angular change of the residual stream between adjacent layers, the authors automatically identify “knee points” that delineate encoder and decoder boundaries. The middle block is then recursively executed multiple times during inference.

**Strengths:**

* The experimental setup is clear and easy to follow, with well-defined baselines and ablation choices.
* The paper conducts comprehensive empirical analysis, comparing multiple ETD configurations (e.g., varying iteration counts, layer boundaries, and adaptive depth)
* The benchmark coverage is broad, across factual, commonsense, mathematical, and reasoning categories, which helps demonstrate the generality of the approach

**Weaknesses:**

* The evaluation relies primarily on a single model (OLMo-2 1B). It would strengthen the paper to include results across different model sizes or architectures (e.g., other OLMo sizes) to demonstrate that the Kneedle-based boundary detection generalizes beyond one model.
* In Figure 2, the angular-distance curve appears rather smooth, without a clear “knee.” This make me wonder whether the automatically detected boundary is robust or merely an artifact of one model’s noise pattern.
* The conceptual link between the angular change turning point and the claimed “encode then think” transition is somewhat heuristic. Reduced representational drift does not necessarily imply reasoning onset, more interpretation or insights into this will be appreciated
* It's necessary to also compare with a larger model under the same FLOPs  to show the trade off .

**Questions:**

see weakness

---

> ### Author Response · Authors · 2025-11-21
> **Author's rebuttal (part 1)**
>
> We first would like to thank the reviewers for their feedbacks and comments. Please find below our responses addressing the weaknesses.
>
> ## Reply to Weakness 1:
>
> Thank you for the thoughtful suggestion — we agree that evaluating ETD beyond one a single model is an interesting direction. As the reviewer 8tY2 also asked about our choice of experimental setup, we address this point in the general response to all reviewers.
>
> ## Reply to Weakness 2:
>
> Thank you for raising this point and we assume that the reviewer is referring to Figure 1. We agree that relying solely on visual inspection of the angular-distance curve could make the boundary detection appear sensitive to noise. However, the Kneedle turn-point is not used heuristically; it is validated empirically through downstream performance.
>
> Specifically, Figure 2 evaluates a broad grid of layer partitions around the automatically detected split. The assignments corresponding to the detected turning point (7-4×2-5) achieve the strongest overall reasoning performance among feasible configurations of E-4*2-D in this grid. Table 6 in Appendix E presents the same results in tabular format to make the comparison explicit.
>
> To further strengthen the robustness argument, we are expanding the ablation space beyond the 4-layer recursive block used in the main analysis. We now evaluate recursive block sizes of 2, 4, 6, 8, and 12, which diversifies the candidate turning points and stress-tests the boundary selection under substantially different curve shapes. . To ensure a controlled comparison, we vary the block size by adding/removing layers symmetrically around the originally selected 7–4*2–5 configuration, keeping the recursive block centered in the same region of the model while changing its extent. We provided the figure in updated version of paper (Figure 3) where we plot the performance vs FLOPs.
>
> We can observe that the performance improves with more layers in the recursive  “thinking” block which is expected. The important observation is that for mathematical reasoning even under same FLOP budget, looping more with 7-4*k-5 configuration is better than looping twice over more layers.
> This trend reinforces the central conclusion of the paper: where and how computation is applied matters more than compute alone, and is further supported by the complementary experimental results. We added a new subsection in Section 4 with results of new ablations (highlighted in blue in the revised version).
>
>
> In summary, the  validity of the proposed method to identify the “think” block is supported by systematic ablation across the surrounding configurations.

---

> ### Author Response · Authors · 2025-11-21
> **Author's rebuttal (part 2)**
>
> ## Reply to Weakness 3:
>
> We agree that reduced representational drift does not automatically equate to the onset of reasoning. Our intended interpretation is more modest: the angular-change profile provides a scalable proxy for identifying where the model transitions from rapid information encoding to sparser iterative computation. This proxy is useful for initializing the recurrent slicing, but it is not presented as a standalone cognitive interpretation. The transition point is validated empirically in Figure 2 (or Table 6). We also added new supporting experiments varying the size of recursive block in Figure 3 (or Table 7).
>
> Our approach on measuring angular distance is based on Gromov et al, 2024 (ICLR’25). They discovered that later layers change the direction of hidden representations less than earlier layers. They used the average angular distance as a criterion for identifying layers to prune. Their experiments show that removing such layers has almost no impact on tasks heavily relying on knowledge retrieval. Despite the low average angular change, however, even moderate pruning of those same layers results in a degradation on reasoning tasks.
>
> Taken together, we view angular change as a principled architectural signal that guides the placement of recurrence — and whose usefulness is ultimately validated through task-level performance.

---

> ### Author Response · Authors · 2025-11-21
> **Author's rebuttal (part 3)**
>
> ## Reply to Weakness 4:
>
> Thank you for the suggestion. We agree that a FLOPs-matched comparison can shed light on the trade-off between recurrence and simply scaling depth.
>
> To offer such comparison without introducing confounders, we have run this comparison:
> - 7-4×2-5 (recurrence, looping a 4-layer block twice), and
> - 7-8×1-5 (non-recurrent, doubling depth to match FLOPs, initialized by stacking the same 4-layer block).
> Stacking the same 4-layer block to mimic the two iterations while keeping the FLOPs constant. Both configurations perform identically before mid-training. Both systems use identical data, hyperparameters, and checkpoints. Results are as follows:
>
>
> | Model       | Params / FLOPs | Factual Knowledge | Reading Comprehension | Commonsense Reasoning | Multi-Disciplinary Reasoning | BBH  | Math. Reasoning |
> |-------------|----------------|-------------------|------------------------|------------------------|------------------------------|------|------------------|
> | OLMo-2      | 16 / 16        | 37.55             | 52.19                  | 65.29                  | 45.00                        | 31.80 | 24.31           |
> | 7-8-5       | 20 / 20        | 31.78             | 52.03                  | 62.45                  | 44.42                        | 30.21 | 21.68           |
> | **7-4×2-5** | 16 / 20        | **38.10**         | **56.14**              | **66.74**              | **48.41**                    | **31.67** | **28.27** |
>
> Key property of ETD is that it leverages the pretrained model efficiently.Results in table above show that the larger iso-FLOPs model underperforms both the original non-recurrent baseline and the ETD (k=2) model, indicating that simply increasing effective depth does not reproduce the improvements achieved by ETD.
>
> We added a new subsection in Section 4 with results of new ablations (highlighted in blue in the revised version).

---

> > ### Comment · Reviewer_Qf7A · 2025-11-28
> >
> > thank the authors for their response. I think my current score is reasonable and will keep my score.

---

> > > ### Author Response · Authors · 2025-12-02
> > > **Summary Reply from authors**
> > >
> > > We thank the reviewer for the thoughtful feedback, which we found helpful in strengthening our contribution.
> > > All raised weaknesses have now been clarified or addressed in the revised manuscript:
> > >
> > >  1. We clarified the scope of the 1B setup and added new results on additional larger model;
> > >
> > > 2. We strengthened the validation of the Kneedle-based split through an expanded grid of ablations and new results varying the recursive block size;
> > >
> > > 3. We clarified the interpretation of angular distance to reflect an empirically validated architectural signal rather than a cognitive claim;
> > >
> > > 4. We added a FLOPs-matched comparison showing that simply increasing depth does not reproduce the gains of ETD.
> > >
> > > We believe the updated manuscript now reflects the correct scope of claims and provides substantially stronger empirical support for the method.

---

### Official Review · Reviewer_H7YH · 2025-10-31

**Soundness:** 2
**Presentation:** 2
**Contribution:** 3
**Rating:** 6
**Confidence:** 4

**Summary:**

This work presents a practical methodology for turning fixed depth transformer LLMs in to recurrent transformers with the specific goal of enhancing reasoning performance without requiring additional trainable parameters. They propose an algorithmic method for choosing how to assign layers from the original model into a three phase encode, think, decode segmenting of the layers in the recurrent model and ablate their design choices. They also explore an adaptive exiting method for also assigning a variable amount of compute per token. Their strong results on mathematical and reasoning intensive benchmarks underscore the promise of recurrent approaches for efficient reasoning.

**Strengths:**

1. Experiments are performed on models at the now ubiquitous 1B parameter scale for open source models, implying their results could directly translate to open source resource constrained applications.
2. They identify an important problem in the adaptation process for turning a fixed depth model into a recurrent one and apply both recent results from the LLM literature and a more classical technique from curvature analysis to propose an automated process for slicing up the model layers optimally when initializing their new model.
3. They report non trivial improvements in reasoning performance over the fixed depth baseline they start with (given the limited absolute capability of 1B models)

**Weaknesses:**

Overall, the paper is rated very borderline on account of some of the ablation soundness and presentation clarity issues discussed below. As the available middle ratings this cycle are just 4 or 6, I am happy to lean very weakly in the direction of acceptance, however, the demerits are represented in the individual component scores above.

### 1. More limited novelty in ETD structure than claimed

Note that there is a bit of a is/ought distinction to be made about where to draw the line between encode think and decode layers. This first should be clarified in S1, and then it also factors into how ablations are performed in S4.3 against other recurrent layer assignment strategies.

Geiping uses a "Prelude, Recurrence, Coda" structure and Bai proposes a "Middle-Cycle" recurrence strategy, which are both essentially the same as the "Encode, Think, Decode" setup proposed in this work. They key thing to note about whether or not a design is "ad hoc" or "optimal" here likely hinges on the training setting. Geiping trains their models from scratch with encode=prelude, think=recurrence, and decode=coda separations defined from the beginning of training and thus one would assume that by the end of training the "roles" of these layers have become aligned with their usage.
In this work, you instead start with a _pretrained_ fixed depth model whose layers have already implicitly come equipped with "roles" as a function of where they were in the original model during pretraining. Hence, one imagines there is likely an optimal way in which these layers could be partitioned when the model is adapted from fixed depth to recurrent; your work studies one such strategy. But the point is that whether or not there exists an optimal partitioning, and whether this one is the one, is more an artifact of the adaptation setting this work studies rather than a true difference between this proposal and architectures in recent work on recurrent llms.

### 2. More limited evidence for optimality of layer division strategy than claimed

The evidence for the use of the Kneedle algorithm is weak and overall the draft space allocated to the optimality motivations in S 2.1 and the design of the S4.3 ablation doesn't support a strong claim that this work has identified a procedure that should generalize to any other model family or experimental setting. Please link Appendix E early on to indicate that _an_ ablation was performed to back the Kneedle based choice, but in S4.4 it would also be helpful to motivate why the recursive block was fixed to a size of 4 before this ablation was performed.

The reader is still left wondering whether the 7-4\*k-5 splitting was actually optimal (even for just Olmo 2). We are missing more targeted experiments where another splitting is used based on a different criteria than the kneedle algorithm, except for the two comparisons in Table 4 against a 2-12\*2-2 and 0-16\*2-0 non recurrent setup. Related to the is/ought comment about the difference between a from-scratch and a pretrained setting above, the way this ablation is set up these are not even weak evidences in support of the optimality argument or use of the kneedle algorithm over a visual check of Figure 1.

Instead the two comparisons are just two more extreme and suboptimal choices. One can interpret the results as 2-12\*2-2 allocating too many layers (from this pretrained model) to the recurrence and not enough to the encoder or decoder, the 0-16\*2-0 results can be viewed as an extreme even more obviously poor choice based on intuitions and prior work regarding the special role of embedding and unembedding layers. Point being, this ablation does not even answer a simple question like whether or not a small change like allocating 5 layers to the thinking stage rather than 4 would outperform the reportedly optimal setup that uses a fancy splitting algorithm. Based on Fig 1 right, it might perform nearly the same.


### 3. Lacking clarity in ACT section

Training details for S5.1 describing the ACT method are missing. How is the router supervision performed? eg. what labels are used for the optimal depth per token position? As this is similar to any router problem such the MoEs expert selection, a non-differentiable choice is made when an exit occurs because no other routes (more iterations of think block in this case) are considered other than the one selected by top-k or argmax. This issue normally requires applying a straight through estimator for the router function to make the process trainable end to end. Additional details about exact what loss was used in these exiting experiments are required.

**Questions:**

1. S4.1 and S 4.2 have similar titles and read very similarly. Essentially they discuss the same series of results on the effect of depth k but this isn't clear on first skim. They should be unified into a single section perhaps with bold paragraph titles that discuss Table 2, and then the breakdown in Table 3, and then maybe point ahead to the appendix (if that's where they are) discussing trends in various individual tasks, but I think this last part could be omitted or moved to just accompany wherever the table is that contains the full task breakdowns.

2.  L45 seems to be an unfinished sentence

---

> ### Author Response · Authors · 2025-11-21
> **Author's rebuttal (part 1)**
>
> Thank you for your valuable feedback!
>
> We would like to address the weaknesses below.
>
> ## W1: More limited novelty in ETD structure than claimed
>
> We thank the reviewer for this thoughtful comment and the opportunity to clarify the novelty of ETD relative to concurrent recurrent designs such as Prelude–Recurrence–Coda (Geiping et al., 2025) and Middle-Cycle (Bae et al., 2025). At a high level, we agree that these works and ours share a three-phase conceptual framing. However, this similarity is superficial: *the adaptation setting fundamentally changes the requirements* and therefore motivates methodological contributions that are absent in prior work.
>
> Unlike other proposals, the idea of ETD is to capitalize on the extensive resources sunk into regular non-recursive models, which are trained anyways. The adaptation setting presents unique challenges that require novel solutions:
> - Pre-existing Layer Roles: Pretrained models already exhibit differentiated layer functions, changing the optimization problem compared to training from scratch. Partitioning the model naively generally disrupts these functions.
> - Compatibility Requirements: ETD preserves the original architecture and requires no additional parameters (e.g., no input injections, LoRA modules, or structural changes), enabling practical deployment on existing state-of-the-art pretrained models such as OLMo-2.
>
> To address this adaptation-specific problem, ETD introduces an interpretability-driven method that infers segment boundaries from representational evolution (angular drift) across layers. To the best of our knowledge, this is not present in prior recurrent LLM designs, which rely on heuristic or manually chosen boundaries. Empirically, Table 4 shows that non-interpretable allocation strategies perform substantially worse in the adaptation regime, confirming that layer partitioning is not interchangeable and does not “automatically align” when adapting pretrained models.
>
> Finally, although ETD is designed for pretrained adaptation, we believe these findings can also inform training-from-scratch setups: if architectural segment boundaries learned during long pretraining implicitly reflect the emergence of roles in layers, then ETD may serve as a strong prior for future recurrent architectures. We also note that prior work has not analyzed layer partitioning in the training-from-scratch regime.

---

> ### Author Response · Authors · 2025-11-21
> **Author's rebuttal (part 2)**
>
> ## W2 - More limited evidence for optimality of layer division strategy than claimed
>
> We agree that stronger empirical support for the optimality of the ETD layer division strategy is important, and we appreciate the opportunity to clarify what is already evaluated and to expand the experiments accordingly.
>
> Table 4 and Figure 2 show different ablations of E-Txk-D configuration. Figure 2 and Table 6 in Appendix E are the same results presented in different formats (graph vs table). Appendix E was initially mentioned in the footnote of section 4.4 and we moved it to the main text (see update file blue-colored text). Experiments in Table 4 compare with alternative recursive frameworks from prior works, and ablation in Figure 2 systematically varies the layer partition across a broad search space of E-4x2-D assignments and demonstrates that the 7-4x2-5 split achieves the highest reasoning performance among feasible configurations. Hence under the same computation budget, the position and composition of the recursive block are the main drivers of performance on reasoning tasks.
>
> We acknowledge that additional ablations with varying recursive block size would complete the picture, and thank the reviewer for the suggestion.  We present additional results with recursive block size of 2, 6, and 8, so overall we ablate with different sizes of 2, 4, 6, 8, 12. To ensure a controlled comparison, we vary the block size by adding/removing layers symmetrically around the originally selected 7–4x2–5 configuration, keeping the recursive block centered in the same region of the model while changing its extent. We provided the figure in the updated version of paper (Figure 3) where we plot the performance vs FLOPs.
>
> We can observe that the performance improves with more layers in the recursive  “thinking” block which is expected. The important observation is that for mathematical reasoning even under same FLOP budget, looping more with 7-4xk-5 configuration is better than looping twice over more layers.
> This trend reinforces the central conclusion of the paper: where and how computation is applied matters more than compute alone, and is further supported by the complementary experimental results.
>
> We added a new subsection in Section 4 with results of new ablations (highlighted in blue in the revised version).

---

> ### Author Response · Authors · 2025-11-21
> **Author's rebuttal (part 1)**
>
> ## W3: Lacking clarity in ACT section
>
> We agree that the submitted draft did not sufficiently clarify how the ACT router is trained. We have now expanded Section 5.1 with the necessary details.
>
> In adaptive-depth ETD, no auxiliary losses, reinforcement learning, new labels, or handcrafted “optimal depth” targets are used. The router is trained purely through the standard task loss, backpropagated end-to-end through all unrolled iterations of the recursive block. Since halting is determined by the accumulated continuous halting probability $H_{t}$​, the router remains fully differentiable during training, and therefore no straight-through estimator is required.
>
> The halting probability $w_{t}$ is produced by a linear projection of $h_{t}$ followed by a sigmoid, and gradients flow both to the router and to the recursive block T through the unrolled computation graph. This does not involve non-differentiable top-k or argmax routing as in MoE architectures, which is why MoE-style straight-through estimators are unnecessary. We now emphasize this distinction explicitly.
>
> To ensure clarity, we also added the following sentence in Section 5.1 (highlighted in blue in the updated version):
>
> “
> Adaptive-depth ETD is trained end-to-end in the same setting as fixed-depth ETD; no new labels, supervision signals, auxiliary losses, hyperparameter tuning, or ad-hoc solutions adopted when training routers for MoE models (Cai et al., 2024; Mu & Lin, 2025) are introduced.
> ”

---

> ### Comment · Reviewer_H7YH · 2025-11-23
>
> I appreciate the author's thorough response to my review and their efforts to update the manuscript accordingly. I have a few additional comments and questions.
>
> ### Lack of evidence to support the optimality of angular distance + kneedle method.
>
> While there is some subjectivity here, despite the authors' attempt to reiterate how they see their approach as a fundamentally more principled one then prior work, the phrasing in L145 , i.e. calling other methods "ad hoc", is still too strong. Presenting a theory for optimal layer partitioning (angular displacement and kneedle) does not necessarily make the method more sound than a grid search likely employed in prior work. One must show that the predictive model identifies an optimal solution out of a reasonable set of other possible configurations, and the experiments and ablations presented are insufficient for this kind of claim.
>
> Specifically, I understand the computational cost of a more complete ablation, but unfortunately Figure 3 does not conclusively show that among possible alternates at a given flop budget, the solution identified by the kneedle method ($7-4\*k-5$) is actually optimal. The main issue is that there are only single alternate choices considered at any specific flop budget. Often $7-4\*k-5$ is equivalent or better, but not always. There is also a claim made in the response text but not S4.4 text (hard to see what is included in one place versus the other) that "performance improves with more layers in the recursive 'thinking' block" even though the 12 layer configuration often performs worse than the 8 layer.
>
> So, in addition to the sparse sampling of one alternate per flop target shown, it would be a bit more convincing if for at least one flop target, a collection of alternate configurations were tested, i.e. a stack of vertical points. Overall, there just isn't sufficient evidence that this angle+kneedle approach that much more effective than a grid search even though it is presented as being much more principled.
>
> ### Remaining details on the training setup for the ACT scheme
>
> I appreciate the added explanation regarding the ACT implementation but I still am unclear on the exact training process. The response states that all unrolled iterations of the recursive block are considered in the training step. This suggests that for each depth value k, the hidden states at that layer are passed through both the router to produce $w_t$ and are passed through the decoder block to produce token predictions for which individual losses can then be computed, lets call that $l_t$. If we consider say $k \in [1,5]$ then there would be $k$ distinct sets of output logits from the model which would produce losses $l_t$ and then we would backpropagate the token prediction errors made at each of the k exits through their respective subset of the total computation graph. If this description is correct, then I am wondering how the $w_t$ values (or their aggregate $H_t$) are actually supervised such that the router will be trained? The way I have described it, $w_t$/$H_t$ are not part of the loss so there is nothing to supervise the linear projection plus sigmoid branch.
>
> I was thinking about the details above a bit while looking more carefully at Fig. 4, but then actually started to wonder why the results look like they do for the adaptive-ETD scheme. For 2 of the 6 tasks, the adaptive exit performs better than the best result for the deepest fixed depth scheme (Drop, OBQA), for 1 of the 6 it performs better than the equivalent depth fixed variant (AGIEval), and then for the other 3 of 6 it performs both worse than its equivalent depth and worse than the deepest fixed setting (GSM,MMLU,ARC). From these results, it really just seems as if the adaptive depth scheme doesn't work very well? One type of "success" you might expect would be that the adaptive method performs equivalently to the fixed depth scheme corresponding to the average even though the deepest fixed versions do better. Or another outcome might be that the average depth the method uses at test time ends up being relatively high (say between 4 and 5), but that the method at least recovers the most of the performance of the fixed depth scheme at k=5. But we don't see either of those.
>
> Is it possible to take the samples behind the star in Fig. 4, and group them by the automatically chosen evaluation depth for each question, and plot the depth wise accuracies? Seeing the exit distribution per task, and performance marginalized by the k values, might start to help explain why the performance of the adaptive method (at least plotted at its average) seems so unsystematic. Spitballing here, given that the average depth in Fig.4 is around 3, is there some argument based on the random router weight init, sigmoid, and eps value, that is expected to produce an expected t value of about 3? Alternately perhaps convergence in states $h_t$ somehow implies by construction that $H_t$ approaches $1-\epsilon$?.

---

> > ### Author Response · Authors · 2025-11-24
> > **Author's reply to Comment (part 2)**
> >
> > ## On the ACT details.
> >
> > Thank you for probing further. We understand that the submitted draft did not sufficiently communicate how gradients flow in the adaptive-depth variant. We will improve Section 5.1 by adding the following details:
> > - At the end of each iteration, the output of the T block goes to the router, which returns the halting value $w_{t}$. $w_{t}$ is then added to $H_{t}$. If $H_{t}$ is less than 1-epsilon, we continue to iterate, otherwise the representation goes to the final layer in D block and is unembedded. The router is only used for making decisions to continue iterating or not. We do not use all outputs of a recursive block, but only the output after the final iteration, when computing final loss. So, adaptive-depth ETD, i.e. the ACT setup, was trained using the same loss as non-recurrent and fixed-depth ETD model.
> >
> > We present our current results as "exploratory approach in the direction of adaptive test-time computation" and we acknowledge that more work needs to be done. We provide some observations in the current ACT setup, but we plan to continue working towards the results you have mentioned: either showing that adaptive depth matches the performance of the fixed-depth model that uses the same average number of iterations (even if the deepest fixed variant still performs best), or demonstrating that the learned average depth at test time is relatively high (e.g., 4–5) while still recovering most of the performance of the fixed-depth model.
> >
> > We appreciate the reviewer’s continued constructive feedback, and we will adjust the manuscript to ensure that the claims and evidence are fully aligned. We believe that the updated writing will more accurately reflect the technical contribution of ETD and reduce the risk of misinterpretation.

---

> > > ### Comment · Reviewer_H7YH · 2025-11-26
> > > **Cont'd discussion (1/2)**
> > >
> > > I appreciate the authors' continued engagement with the comments and concerns raised in my original review.
> > >
> > > While Sec 4 remains, in my opinion somewhat incomplete, I understand that these experiments are costly. So if the authors can include a sentence similar to "the Kneedle-based partition selects a split that lies near the performance maximum in the search space across tasks" at the end of Sec 2.1 where the selection algorithm is discussed (eg. lines L178-L181) and also somewhere in Sec 4, then I think that the claims and experiments will be in better balance.

---

> > > ### Comment · Reviewer_H7YH · 2025-11-26
> > > **Cont'd discussion (2/2)**
> > >
> > > Unfortunately, I continue to have an incomplete understanding of the ACT section despite the authors' continued reiteration of the basic point that "Adaptive-depth ETD is trained end-to-end in the same setting as fixed-depth ETD". Note that since no straight through estimation is used for router training, you don't need to mention MoEs even though I mentioned this in my original review; including these citations now simply clouds the picture as it alludes to things that are completely irrelevant for your method (finding this out was the point of my original questioning).
> > >
> > > I will attempt one more time to describe my understanding of this approach and see if my misunderstanding can be resolved. If not, I recommend that this part of the draft be removed from the manuscript as it potentially suggests that the method admits an test time compute adaptivity that it actually does not, and therefore leaving it in would detract from the soundness and transparency of the work.
> > >
> > > Quoting the authors above (this blurb is not actually in updated PDF yet I think):
> > > > "At the end of each iteration, the output of the T block goes to the router, which returns the halting value $w_{t}$. $w_{t}$ is then added to $H_{t}$. If $H_{t}$ is less than 1-epsilon, we continue to iterate, otherwise the representation goes to the final layer in D block and is unembedded. The router is only used for making decisions to continue iterating or not. We do not use all outputs of a recursive block, but only the output after the final iteration, when computing final loss. So, adaptive-depth ETD, i.e. the ACT setup, was trained using the same loss as non-recurrent and fixed-depth ETD model."
> > >
> > > What this suggests is that **this is purely a test-time adaptive exit method; no modification to the training process or architecture is done.**
> > >
> > > Below I sketch a vague rewrite of the ACT section that might describe what is going on more clearly.
> > >
> > > ---
> > > The proposed ACT procedure is implemented as follows:
> > >
> > > 1) Train the ETD model at the fixed maximum depth that is desired at test time, say $k=5$ for example.
> > >
> > > 2) At test time, randomly initialize a linear projection matrix to be applied to the hidden states output from the recurrent block; the shape of this projection matrix is **(<what?>,1)**; eg. the transformation is applied position wise independently to all hidden state vectors, and the same transformation matrix is used for all repetitions of the repeated thinking block.
> > >
> > > 3) Then, during generation, the exit signal is the composition of the linear transformation applied to each layer with a sigmoid and those scores are aggregated via Eq. 3 to yield $H_t$.
> > >
> > > 4) For each generated token, once the aggregated score $H_t$ passes a threshold $1-\epsilon$, where for example epsilon is chosen to be **<what?>** based on **<what?>**, then the hidden states of the layer are passed to the decoder block and a new token is generated.
> > >
> > > 5) The adaptive depth at which the router caused the model to exit is summarized by the average over all tokens in a rollout, and then averaged over all samples in a benchmark as reported in Fig. 4.
> > >
> > > A critical remark is that the router matrix is initialized as a random projection and is left untrained. Therefore, the exit depth, for a given distribution of hidden states as input (say a specific benchmark domain like GSM), is directly modulated by the value of epsilon chosen. The rate at which $H_t$ approaches $1-\epsilon$ is a function of the natural hidden state magnitudes produced by the model at each layer, and varies as a function of the particular prefix and token input at a given position. However, this dependence is hard to analyze without additional knowledge about the expected distribution of hidden states, and critically, _is not optimized for in any way_. **While one could train the router matrix to cause the accumulated score $H_t$ to pass the threshold at approximately the minimum depth required to produce the correct token/yield high reward, developing this kind of learned ACT scheme for the ETD method is left to future work.**
> > >
> > > As a result, Fig. 4 shows that given a fixed epsilon value of **<what?>**, for all benchmarks considered, on average $H_t$ surpasses the exit threshold at between 3 to 4 iterations. For each benchmark, a histogram of the exit depths are show in Appendix Fig. **<what?>**.
> > > Alternate training-free schemes inspired by prior work on recurrent models and diffusion language models would include checking for convergence in hidden states, eg. $sim(h_t-1, h_t)$, as a function of depth, or by applying the decode block at each candidate depth and tracking the predicted token confidences, but these are also left to future work.
> > >
> > > ---
> > >
> > > I remain unsure about whether I understand the ACT section or whether it even makes sense, so I would appreciate it if the authors are able to try and confirm whether my updated understanding of the approach is correct.

---

> ### Author Response · Authors · 2025-11-24
> **Author's reply to Comment (part 1)**
>
> We sincerely thank the reviewer for the continued engagement and the detailed follow-up comments. We understand the concerns raised, and based on the feedback, we will update the manuscript to more accurately align our claims with the current level of evidence.
>
> ## On the optimality claim of the angular-distance + Kneedle split.
> Your comments draw our attention to the fact that we have used the adjective ‘optimal’ more liberally than we should have, and we thank you for that.
> To clarify an important point, to our knowledge, prior works that partition layers in three groups do not identify a split through grid search, which would be quadratic in the number of layers and computationally prohibitive to most: they each consider a single split for each model scale. ETD’s core contribution is a method, grounded in previous results, to determine a split that is empirically effective, especially for reasoning-heavy tasks.
>
> In the camera-ready version, we will revise the wording to reflect these considerations:
> - We will replace the statements that previous works adopt ad-hoc solutions with a clarification that they consider a single partition of layers into three groups, without exploring alternatives, which is what we meant.
> - Rather than implying optimality, we will report that the Kneedle-based partition selects a split that lies near the performance maximum in the search space across tasks.
>
> We experimented with configurations along two dimensions to support that 7-4xk-5 is the preferred configuration for OLMo-2 1B model.
> - First, we selected 4 layers for T block in different positions: 1-4x2-11, 3-4x2-9, 5-4x2-7, 7-4x2-5 , 9-4x2- 3, 11-4x2-1. These experiments show that the performance is coming not only for additional FLOPs but on choosing relevant layers.
> - Then, upon the reviewer’s request, we ran experiments varying the size of T: 0-16x2-0, 2-12x2-2, 5-8x2-3, 6-6x2-3, 7-4x2-5 , 8-2x2-6. The reviewer noted that, as more layers are included in the recursive “thinking” block, the performance improves only until some point: the 12 layer configuration often performs worse than the 8 layer. This provides additional support for the claim that carefully selected layers are important and not just additional FLOPs. We will change the claim to reflect this observation.
>
> We agree more ablation studies could explore further dimensions. However, considering the computational cost of running different configurations of the 1B parameter model (see table below), as the reviewer highlighted, it would be difficult to do this in time for the end of the discussion period. Nevertheless, we believe our results provide insights on designing ETD-like recursive architectures that are worth sharing with the research community.
>
> Table below reports the training time of experiments presented in Table 2. We run all experiments on a node with 8 A100 80GB GPUs.
>
> | Model       | GPUs        | Hours / run | GPU hours |
> |-------------|-------------|-------------|-----------|
> | OLMo2 (k=1) | 8 × A100     | ~116        | ~928      |
> | ETD (k=2)   | 8 × A100     | ~137        | ~1,096    |
> | ETD (k=3)   | 8 × A100     | ~170        | ~1,360    |
> | ETD (k=4)   | 8 × A100     | ~195        | ~1,560    |
> | ETD (k=5)   | 8 × A100     | ~220        | ~1,760    |

---

> ### Author Response · Authors · 2025-11-27
> **Author's Reply to Cont'd discussion**
>
> We sincerely thank the reviewer for the continued engagement throughout the discussion period. Your feedback has been instrumental in improving both the clarity and balance of the manuscript.
>
> ## On the angular-distance + Kneedle split.
> Following your suggestion, we revised the text in both Section 2.1 and Section 4.1 (highlighted in blue) to accurately reflect the current scope of evidence and place the method on the right level of claims. We believe that this brings the claims and experiments into the balance you intended and improves the clarity of the contribution for readers.
>
> ## On the ACT details.
> Thank you for making the effort to articulate in detail **your understanding of the adaptive-depth extension of the method: it is not correct**.
> - The exit router is trained together with the rest of the model during mid-training stage
> - The exit test is performed only at the end of a full T block, not after each layer
>
> Our adaptive-depth ETD extends the fixed-depth ETD formulation by allowing the model to determine the number of recursive iterations per token.
>
> On the architecture:
> - We keep the general architecture of the model the same and add a lightweight router.
> - The router is implemented as a linear projection of the hidden state followed by a sigmoid activation.
> - The input to the router is the hidden representation that is output by the recursive T block, and the output of the router is the halting value between 0 and 1
> - The router is randomly initialized.
>
> At training stage:
> - Adaptive-depth ETD is integrated into mid-training in the same way as fixed-depth ETD.
> - We train the router to learn how to allocate resources, i.e. iterations, for different input tokens, at the same time as we mid-train the other model parameters
> - Early during training the router may output extremely small halting values, causing excessively many iterations. To avoid - this, we cap the maximum number of iterations during training to Nmax=10.
> - We do not provide auxiliary losses (e.g., intermediate losses after each iteration) nor do we introduce any regularizers.
> - Hyperparameters—including optimizer, learning rate, and scheduler—remain identical to fixed-depth ETD.
>
> At test-time:
> - The test time regime is very similar to the training regime, except that once the model is trained we remove the cap on the number of iterations.
> - The model determines on its own the number of iterations: after each iteration the router uses the output of the recursive block to predict the halting value for the iteration, and stops as soon as the cumulated halting values exceed 1-$\epsilon$ ($\epsilon$=0.01)
>
> Full architectural, training, and inference details are now provided in Appendix D titled “Details on Adaptive-depth ETD training”.

---

### Author Response · Authors · 2025-11-21
**General reply to reviewers**

We thank all reviewers for valuable feedback.

Some reviewers (Qf7A, cBrU, 8tY2) showed interest in results of ETD on other models and scale. We appreciate the value of evaluating ETD across different model families.

Our choice of OLMo-2 1B was intentional: it provides a controlled environment where we can isolate the effect of ETD without adding confounding factors. ETD integrates into the training pipeline rather than as a finetuning adapter, and attributing performance changes exclusively to recurrence requires access to:
- intermediate and final checkpoints,
- the full training data mixture,
- the exact optimizer and hyperparameters.

To the best of our knowledge, OLMo-2 is currently the only open-source model family that exposes all of these components and allows modifying the training loop while holding data, compute, and optimization strictly constant. This property enables us to determine whether ETD alone is responsible for the observed gains, without confounding effects from additional finetuning exposure.

However, we fully agree that testing ETD beyond a single model strengthens generality. To address this, we have extended our experiments to OLMo-2 7B, to the best of our knowledge, we present the largest model trained with a recurrent architecture. The results on 7B follows the same trend: ETD approach improves the model performance on reasoning-intensive tasks such as math as seen below


| Model        | Params / FLOPs | Factual Knowledge | Reading Comprehension | Commonsense Reasoning | Multi-Disciplinary Reasoning | BBH   | Math. Reasoning |
|--------------|----------------|-------------------|------------------------|------------------------|------------------------------|-------|------------------|
| OLMo-2 7B    | 32 / 32        | 56.63             | 74.68                  | 76.73                  | 62.90                        | 48.18 | 41.63           |
| **16-10×2-6** | 32 / 40        | **56.89**         | **75.05**              | **76.82**              | **62.95**                    | **49.77** | **42.64** |

Results on GSM*k and MATH:

| Model        | Params / FLOPs | GSM8K Acc. | GSM8K Δ (%) | MATH Acc. | MATH Δ (%) |
|--------------|----------------|------------|-------------|-----------|------------|
| OLMo-2 7B (k=1) | 32 / 32        | 66.18      | –           | 17.07      | –          |
| **16-10×2-6** | 32 / 42        | **67.02**  | **+1.29%** | **18.26**  | **+6.38%** |



We also want to note that mid-training of both 1B and 7B models used the same amount of data, meaning that 1B was exposed to more data per parameter. We believe that with more training the impact of ETD approach on 7B model will be larger. The training perplexity plot suggests that the model did not plateau yet.


Finally, we note that applying ETD via finetuning on additional data can introduce data-overlap bias: if the finetuning dataset overlaps differently with the pretraining corpora of different models, improvements become entangled with differences in prior data exposure rather than with recurrence. This effect has been observed empirically by Bae et al. (2024), where a non-recurrent baseline lost performance under the same finetuning data—making it difficult to attribute subsequent improvements conclusively to architectural changes.

Bae et al. (ICLR, 2025 )https://arxiv.org/abs/2410.20672

---

### Author Response · Authors · 2025-12-02
**Summary of Revisions and Reviewer Consensus**

Dear ACs,

We thank all reviewers for their feedback. We are delighted to read that reviewers find that:
- Our approach addresses an important and practical problem — converting fixed-depth transformers into recurrent ones — with results that can directly translate to resource-constrained open-source deployments (H7YH, 8tY2).
- ETD delivers non-trivial performance improvements, especially on mathematical and reasoning-centric benchmarks such as GSM8K and MATH, and improves OLMo-2 1B after mid-training (H7YH, cBrU, 8tY2).
- The method offers a simple yet effective recipe for recurrence, including an interpretable criterion for selecting encoder/think/decoder boundaries that is empirically validated (cBrU, 8tY2).
- The experimental setup is clear and easy to follow, with well-defined baselines, ablations, and multiple recurrent configurations (e.g., iteration counts, boundary positions, adaptive depth) (Qf7A).
- The empirical evaluation is comprehensive, covering factual, commonsense, mathematical, and multi-step reasoning tasks, demonstrating the generality of the approach (Qf7A).
- The proposed adaptive-depth mechanism, which determines the number of iterations per input dynamically, is effective and further improves performance–compute trade-offs (cBrU).

Please find below a summary of the shared points raised during the review process and the specific experiments and revisions we introduced to address them.

### **1. On optimality of ETD approach.**

Reviewers questioned that our ETD approach can lead to optimal partitioning of layers.

**Our Response**: We clarified the motivations of using our approach, provided new experimental results, and updated the paper.

- **Clarification and Paper Update**: We clarified that ETD selects a split that performs near the maximum across tasks. Prior works select one split per model scale — making ETD the first to systematically analyze how the choice of split affects performance. We clarified in the paper.

- **New Experiments**: We added experiments varying block size to demonstrate the effectiveness of the configuration chosen by our methodology. See discussion with reviewer H7YH and Qf7A for more details.

### **2. On using a single model.**

Reviewers expressed concerns about using a single model, OLMo-2 1B base, in our experiments.

**Our response**: We explained the rationale for using the OLMo-2 model: it provides a controlled environment where we can isolate the effect of ETD without confounding factors. OLMo is currently the only advanced open-source model family that provides checkpoints, data, hyperparameters, and code.

- **New experiments**: To address this concern, we integrated ETD into the mid-training of OLMo-2 7B and present results in Sec. 4.6.

### **3. On details of Adaptive-depth ETD**

Two reviewers (H7YH, 8tY2) asked for more details in Sec. 5 about adaptive-depth ETD.

- **Paper Update:** We revised Sec. 5 and added a new section in Appendix with all details on the architecture, training, and inference.

### **4. On FLOP-matched non-recursive baseline**

Reviewers Qf7A and 8tY2 expressed interest in comparisons with a FLOP-matched non-recursive baseline in two different setups:
- Setup 1: With the same number of effective layers, N, and same training data, d, so $N_{1}$=$N_{2}$ and $d_{1}$ = $d_{2}$
- Setup 2: With different combinations of effective layers and training data, but with the same product (number of effective layers) x (training data), so $N_{1} \times d_{1}$ = $N_{2} \times d_{2}$.

**New experiments**: We run new experiments and present results for both setups:
- We trained a larger model following Setup 1. See discussion with Reviewer Qf7A for more details.
- We evaluated the recursive model trained on less data to follow Setup 2. See discussion with Reviewer 8tY2 for more details.

The new results highlight the effectiveness of the ETD approach.

### **5. On the robustness of ETD results**

Some reviewers requested to prove the statistical significance of our results.

**Our Response**: It is not standard practice to report standard deviation on mid-training for 1B and 7B models, due to obvious resource constraints. In discussions with reviewers H7YH and 8tY2 we report the training time of main experiments presented in Table 2 in paper.

- **New experiments**: We nevertheless trained the 7-4x2-5 models with two more random seeds. We present standard deviations in the discussion with reviewer 8tY2: the results are very consistent. While ideally statistical significance would be determined on numerous seeds (>3), these results are evidence that differences between ETD and the baselines cannot be ascribed to noise.


### **6. Extension & Clarification of Related Work**

Some reviewers asked to extend the related work with more discussion or clarifications of prior work.

- **Paper Update**: We updated the paper to add additional discussion, related work, and clarifications.

---

### Meta-Review · Area_Chair_x1hS · 2025-12-13

**Summary:**

There are several major concerns raised by the reviewers: 1. In technical principle, the distinction between the proposed method and existing approaches is not clear; 2. Most experiments rely on a single model, limiting the generalizability of the findings; 3. The description of adaptive test-time scaling is unclear. 4. FLOPs matched baselines are missing.

**Reviewer Concerns:**

Regarding the reviewers' concerns summarized above: The authors' rebuttal on distinguishing their method from prior work remains unconvincing. The additional experiment using only the OLMo-2 7B model does not yield significant results and fails to adequately demonstrate generalizability. The additional clarifications on adaptive test-time scaling are still unclear and add confusion. The experimental results on the FLOPs matched baselines are not convincing.

**Reviewer Scores:**

The major concerns raised by reviewers cBrU (4), 8tY2 (2), and H7YH (6) remain unaddressed. Reviewer H7YH initially gave a borderline score of 6 and noted the score could have been a 4. Based on the authors' responses, this reviewer's score is more likely to decrease than increase. Reviewer Qf7A (6) has indicated they will keep their rating unchanged.

---

### Decision · Program_Chairs · 2026-01-26

Reject